

# Investigation of the climatology of low-level jets over North America in a high-resolution WRF simulation

Xiao Ma[1,2], Yanping Li[1,2], Zhenhua Li[1], Fei Huo[1]

[1]Global Institute for Water Security, University of Saskatchewan, 11 Innovation Blvd, Saskatoon, SK, S7N 3H5, Canada

[2]School of Environment and Sustainability, University of Saskatchewan, 117 Science Place, Saskatoon, SK, S7N 5C8, Canada

*Correspondence to*: Yanping Li (yanping.li@usask.ca)

**Abstract.** In this study, we utilized a high-resolution (4 km) convection-permitting Weather Research Forecasting (WRF) simulation spanning a 13-year period (2000-2013) to investigate the climatological features of Low-level Jets (LLJs) over North America. The 4-km simulation enabled us to represent the effects of orography and the underlying surface on the boundary layer winds better. Focusing on the continental US and the adjacent border regions of Canada and Mexico, this study characterizes the spatial distribution, seasonal patterns, and diurnal fluctuations of northerly/southerly LLJ occurrence frequencies. This paper not only identified several well-known large-scale LLJs in North America, such as the southerly Great Plains LLJ and the summer northerly California coastal LLJ, but also the Quebec northerly LLJ, which gets less focus before. Moreover, the high-resolution simulation revealed climatic characteristics of weaker and smaller-scale LLJs or low-level wind maxima in regions with complex terrains, such as the northerly LLJs in the foothill regions of the Rocky Mountains and the Appalachian during the winter. Additionally, the different thermal and dynamic mechanisms forming significant LLJs near the Great Plains, California, and Quebec are investigated. This study provides valuable insights into the climatological features of LLJs in North America and the high-resolution simulation offers a more detailed understanding of LLJ behavior near complex terrains and other smaller-scale features.





## 1.  Introduction

A low-level jet (LLJ) is generally described as the fast-moving air ribbon located in the lower atmosphere (Bonner, 1968; Rife et al., 2010). Many of the world's LLJs have been studied, such as the Great Plains LLJ over the central US (Bonner, 1968; Zhong et al., 1996), the Somali LLJ over eastern Africa (Munday et al., 2021), and the South American LLJ over the east Andes Mountains (Montini et al., 2019). A kind of mesoscale weather system, an LLJ has a relatively small vertical range of usually only a few hundred meters, but its width can reach several hundred kilometers. LLJs are closely related to precipitation and even extreme events, and they can transfer abundant water vapor to the downwind regions, providing favorable dynamic conditions for rainfall (Walters and Winkler, 2001; Hodges and Pu, 2019). Because LLJs also affect processes such as wind power development, air pollution transportation, and urban heat islands (Hu et al., 2013; Sullivan et al. 2017; Gadde and Stevens 2021, Ma et al., 2022), researchers have long been interested in investigating their features.

Since the mid-20th century, scientists have used regular rawinsonde observations to investigate the characteristics of LLJs. Applying rawinsondes to investigate the Great Plains LLJ in the central US, Bonner (1968), Mitchelle et al. (1995), and Walters et al. (2008) studied its distribution, seasonal activity, horizontal and vertical structure, and diurnal features and established the climatology of the Great Plains LLJ during warm seasons. As well as rawinsondes, radar systems and wind profilers are useful tools for characterizing LLJs. Frisch et al. (1992) observed a typical LLJ process using Doppler weather radar in North Dakota and identified that the friction on the surface of the boundary layer is important in the early stages of LLJ development. Using long-term wind profiler measurement, Miao et al. (2018) interpreted the climatology of LLJs in Beijing and Guangzhou, concluding that the frequency values of LLJs in these two cities are 13.0% and 4.9%, respectively. Moreover, Smith et al. (2019) used the Plains Elevated Convection at Night (PECAN) observations to conduct high-quality measurements of nocturnal LLJs with wide spatial and temporal resolutions. They found that sudden changes in LLJ structure typically result from the spatial evolution of the LLJ.

However, there are some disadvantages of observational research that should be noted. First, regular rawinsonde data only contain measurements at two daily time points (00 UTC and 12 UTC), which cannot fully capture LLJs' diurnal variations. The density of observations is therefore coarse, and coastal areas lack regular high-density measurements, making the study of coastal LLJs challenging (Mitchell et al., 1995). Second, heterogeneities in the rawinsonde records, such as variations in station locations, radiosonde types, and archiving procedures, may also complicate the use of





these observations in climate research. Third, rawinsonde measurements taken at a single point are not able to capture
horizontal shear and environmental conditions (Chen et al., 2005). Although observations platforms such as radar or
field projects like PECAN can compensate to some extent for this lack of observational data, these approaches are still
limited by the spatial coverage of their measurement platforms (Smith et al., 2019).
Because of these problems with observational methods, researchers have chosen reanalysis datasets as an alternative
for investigating LLJs. Reanalysis data have relatively better spatial and temporal coverage than rawinsonde
measurements, incorporate observations into the preliminary model simulations, perform more extensive
measurements, and contain broader domains. Rife et al. (2010) highlighted the global distribution of identified
nocturnal LLJs using reanalysis data with a horizontal grid spacing of 40 km, and even successfully extracted some
previously unknown jets. Doubler et al. (2015) applied the North American Regional Reanalysis (NARR) dataset (~32
km) to generate long-term LLJ climatology in North America. Consistent with previous records, Doubler's results
supplemented the description of some smaller-scale LLJs. Similarly, Montini et al. (2019) compared and validated the
performance of five different reanalysis datasets in identifying LLJs. Their results showed the 38-year climatology of
South American LLJs with ERA-Interim data (~79 km).
Scientists have also conducted studies based on numerical simulations, which can more accurately represent LLJs than
reanalysis data sets, especially in the vertical direction, thereby yielding new insights into LLJs' features. Tang et al.
(2017) used an ensemble of dynamically downscaling regional climate simulations to generate the climatology of
Great Plains LLJ and predicted that the LLJ will occur more frequently during the nighttime in spring and summer in
mid-21st century. Jiménez-Sánchez et al. (2019) conducted a simulation for LLJs over the Orinoco River Basin by
dynamic downscaling of the Weather Research and Forecasting model (WRF). The simulation represented the jet
streaks better than previous studies within a broader region of wind enhancement and illustrated more detailed diurnal
evolution. Nevertheless, most general numerical simulations still represent the convective processes by the
parameterization scheme, which generates uncertainty in the results. These issues can be addressed by using
convection-permitting models with grid spacing under 5 km that adequately simulate the convections and other small-
scale processes (Liu et al., 2017, Li et al., 2019, Kurkute et al., 2020). Convection-permitting modeling describes the
underlying surface more accurately than coarse-resolution simulations and reanalysis data and shows promise in
investigations of LLJs near complex mountain areas. Du and Chen (2019) analyzed the LLJs over southern China by



using 4-km WRF model and revealed a solid relationship between the mesoscale lifting of LLJs and the convection's
initiation. They also highlighted the importance of coastal terrain. Overall, the finer-resolution tools tend to show more
comprehensive and precise results, offering detailed and accurate references to LLJs.
The formation mechanisms of LLJs have been studied extensively by researchers. In the inertial oscillation theory
proposed by Blackadar (1957) and Stensrud (1996), it is suggested that the diurnal cycle feature of the Great Plains
LLJ is related to the friction change in the boundary layer. During the night, the jet-core wind is enhanced after
decoupling with near-surface friction. Holton (1967) and Parish (2000) developed the thermal wind adjustment theory,
which suggests that the horizontal pressure gradient changes because the atmosphere over sloping terrain is warmer
or because sea-land contrast influences the diurnal cycle of wind. Additionally, LLJs can also be formed due to
synoptic system forcing, as proposed by Uccellini et al. (1987) and Saulo et al. (2007). However, convection-
permitting models can help explain how LLJs form because they have precise descriptions of weather systems and
underlying orography. Using 4-km simulations, Fu et al. (2018) and Zhang et al. (2019) analyzed the evolution of
LLJs over mountainous areas in eastern and southwestern China, respectively. They concluded that inertial oscillation
plays a prominent role in and is responsible for the local precipitation peak at a certain time. Besides, Shapiro et al.
(2016) argued that the formation of some LLJs may not be impacted by a single factor and that a unified theory
analysis is thus required. Thus, a dataset that offers more information must be very popular. All these studies have
shown that convection-permitting models, with both finer coverage and resolutions, are a powerful tool for LLJ
climatology research.
The purpose of this study is to use the 4-km convection-permitting WRF model (Liu et al., 2017) to produce a detailed
LLJ climatology. This paper focuses on the features of LLJs in major areas of North America and aims to provide
alternative dataset sources with the finer spatial and temporal resolution for the LLJs in this region and provide more
helpful tools for LLJs-related studies in other disciplinary. Section 2 introduces the model configuration and the
criteria for LLJ identification, Section 3 presents the characteristics of LLJ frequencies in North America, and Section
4 illustrates the analysis of the background and mechanisms in several LLJ cases. Finally, Section 5 provides the
discussion and conclusion.



**2.   Model configuration and methods**
**2.1 WRF setup**
This study utilized a convection-permitting Weather Research and Forecasting (WRF) dataset (Liu et al. 2017, Data
available at: https://rda.ucar.edu/datasets/ds612.0/) with a horizontal resolution of 4 km over North America. The
domain covers the entire continental US, Southern Canada, and Northern Mexico, as illustrated in Figure 1. The
simulation provides three-dimensional data at a temporal resolution of 3 hours, resulting in 8-time steps per day. In
the vertical direction, the data have 51 eta levels and can reach 50 hPa. And it should be noted that there are five layers
under 500-m height and nine layers under 1 km are outputted above ground level, which means the WRF has the good
ability to capture the LLJs occurring in the boundary layer. The simulation period spans from 1st October 2000 to
30th September 2013, and the six-hourly ERA-Interim reanalysis dataset of 0.7° resolution was used as input for the
climate simulation. The simulation did not apply any cumulus parameterization scheme due to the fine horizontal grid
spacing, but other sub-grid scale processes were parameterized by various physical schemes: the rapid radiative
transfer model (RRTMG) (Iacono et al., 2008) was used for simulating longwave and shortwave radiations, the Yonsei
University (YSU) scheme was used for representing the planetary boundary layer (Hong et al., 2006), and the Noah-
MP model was used for computing surface processes (Niu et al., 2011). In this study, the planetary boundary layer
scheme is retained, but it should be noted that this would introduce uncertainties to the simulation in the vertical
direction, especially in regions with complex topography.

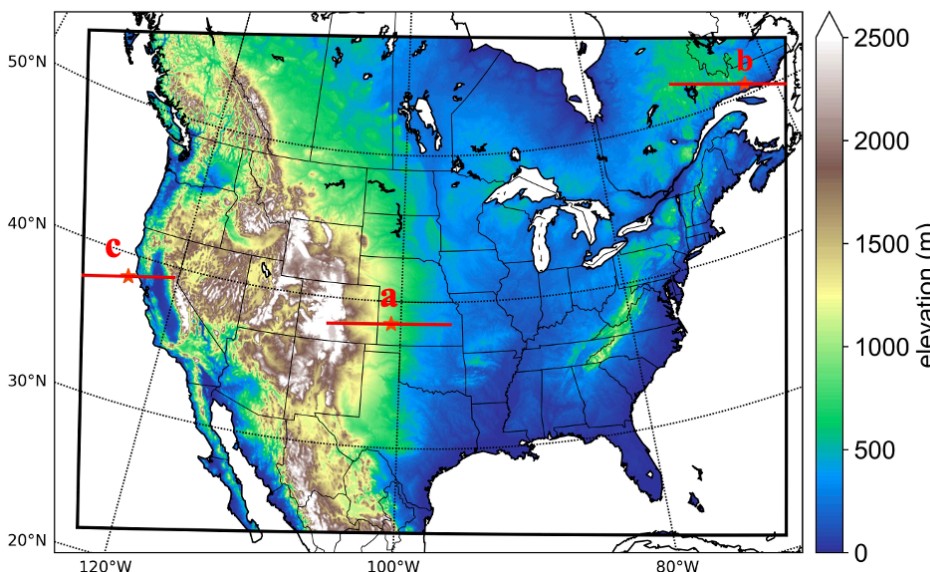

**Figure 1. Study domain of this convection-permitting model. The colors represent the elevation. The red lines and stars show the positions of investigated cross-section and jets in Section 4.**

**2.2 Methodology**

Using the threshold criteria proposed by Bonner (1968), this study identifies LLJs from the vertical wind profile of each grid point in the model output data. LLJs are present when the following conditions are met: (1) the height of the LLJ core maximum wind speed is below 3 km above the ground level (AGL); (2) the maximum wind speed is greater than or equal to 12 m s-1; (3) from the height of the wind maxima to the height of the next minimum value or 3-km height (whichever is lower), the velocity of winds drop by at least 6 m s-1; (4) the wind speed drops by at least 6 m s-1 below the level of wind maxima. Considering the importance of the meridional LLJ for heat and water vapor transport, this study addresses their frequencies in different meridional directions. According to Walter et al. (2008) and Doubler et al. (2015), the criteria for identifying different meridional LLJs are as follows: for southerly LLJs (S-LLJs), the wind direction is between 113° and 247°; for northerly LLJs (N-LLJs), the direction is between 293° and 67°. These criteria are used in this study.

Based on the identification criteria above, we determined if the LLJ existed at each grid point and consequently counted the occurrences of S-LLJs and N-LLJs. We also calculated the frequencies of LLJs in different seasons or



time steps. The frequency is defined as the percentage of the total number of occurrences for the selected accumulation
period. We generated the frequency distribution maps for LLJs climatology in North America, which are illustrated
in Section 3.
**3.    The climatology of North American LLJs**
**3.1 Analysis of atmospheric circulation**
This study adopts model data to capture the climatological features of LLJs in North America. Considering the
relationship between LLJs and synoptical systems, we evaluated the ability of the convection-permitting model to
simulate the background atmospheric circulation. Figure 2 depicts the simulated climatology of geopotential heights
at 500 hPa and sea-level pressure isobars for summer and winter. In summer, at a height of 500 hPa (Figure 2a), In
summer, the model depicts a trough in the east of the continental US, a ridge over the Rocky Mountains, and the
upper-air subtropical anticyclone crossing the southern US. At sea level (Figure 2b), the model captures the Azores
High-Pressure area in the Atlantic Ocean and the Hawaiian High-Pressure area in the Pacific.
In winter, the contours at the pressure value of 500 hPa (Figure 2c) show stronger fluctuating characteristics: the
eastern trough and western ridge over the continent strengthen, and the polar vortex extends to the northern US, while
most of North America is controlled by a cold high-pressure system. In addition, the subtropical anticyclone is too
weak to be found within the study domain. On the other hand, most of North America is controlled by a cold high-
pressure system at sea level (Figure 2d), and parts of the Icelandic Low and Aleutian Low appear on both east and
west of Canada, even though their centers are not captured in the domain. To summarize, the convection-permitting
model can simulate the features of semi-permanent centers of atmospheric circulations in North America, thus
demonstrating its strength in identifying the LLJs in this area.



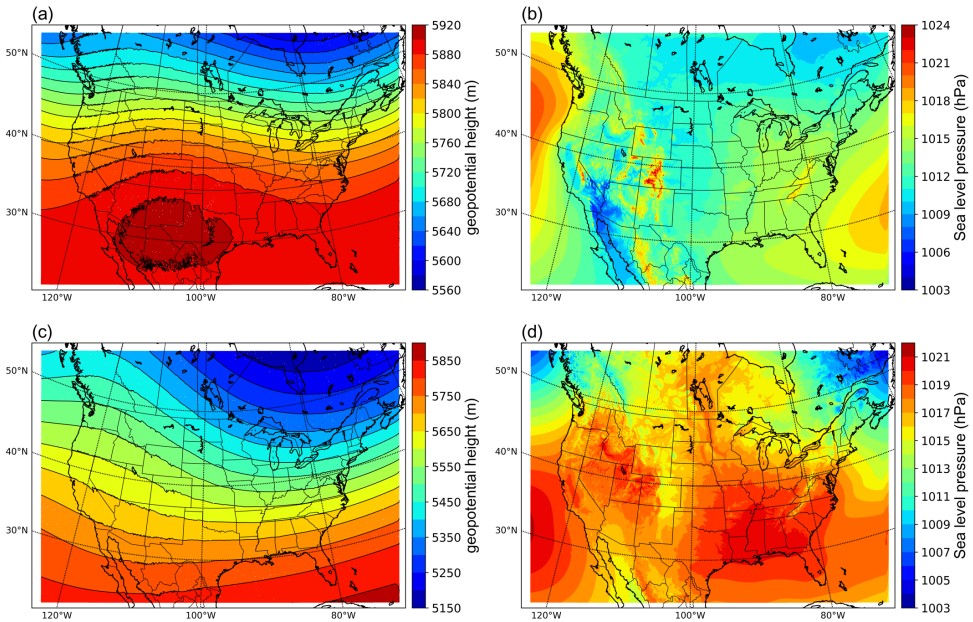

**Figure 2. Climatology of atmospheric circulations simulated by the convection-permitting model: (a) summer 500 hPa geopotential height; (b) sea-level pressure in summer; (c)-(d) the same variables but in winter.**

### 3.2 Seasonal variations of LLJs

### 3.2.1 Northerly LLJs

Figure 3 shows the frequency distribution of N-LLJ in four seasons, in which the frequency represents the ratio between the seasonal total number of LLJs occurrence and the total time steps in each season. Clearly, the California coastal LLJ is strongest in summer (June, July, and August (JJA)), with a large area of N-LLJ frequency greater than 25%, extending from the southern Oregon coast to the central California coast. Regions with a frequency greater than 5% can even extend to the Pacific Ocean near northern Baja California. However, from summer to autumn (September, October, and November (SON)), the frequency of this LLJ decreases sharply, with a frequency of only 5%-15% in the core region, and it is only distributed on the northern coast of California. In winter (Dec, Jan, and Feb (DJF)) it occurs very infrequently (~3%).

On the other hand, various N-LLJ phenomena occur frequently in the cold season. These N-LLJs are mainly located near the eastern slopes of special terrains such as the Rocky Mountains, Appalachian Mountains, and the Quebec





Labrador Plateau. In winter, high frequencies (>10%) are observed from western Alberta to Oklahoma, within which
hot spots are distributed sporadically in Alberta, Montana, Wyoming, and Colorado. These hot spots have frequencies
of about 20%, especially in the region between Colorado and Wyoming. In over 25% of the wind profiles, the N-LLJs
can even be extracted. The N-LLJs over the Eastern US coast mainly extend from Maine to South Carolina, and their
highest frequency can reach about 15%-20%. The N-LLJs in eastern Quebec also occur most frequently in winter
(>25%). Over Hudson Bay, the simulation can also detect the N-LLJ from about 10% of the time steps. The
frequencies of all the N-LLJs mentioned above decline significantly in spring, and it is hard to detect them in summer
as the frequencies are mostly less than 5%.

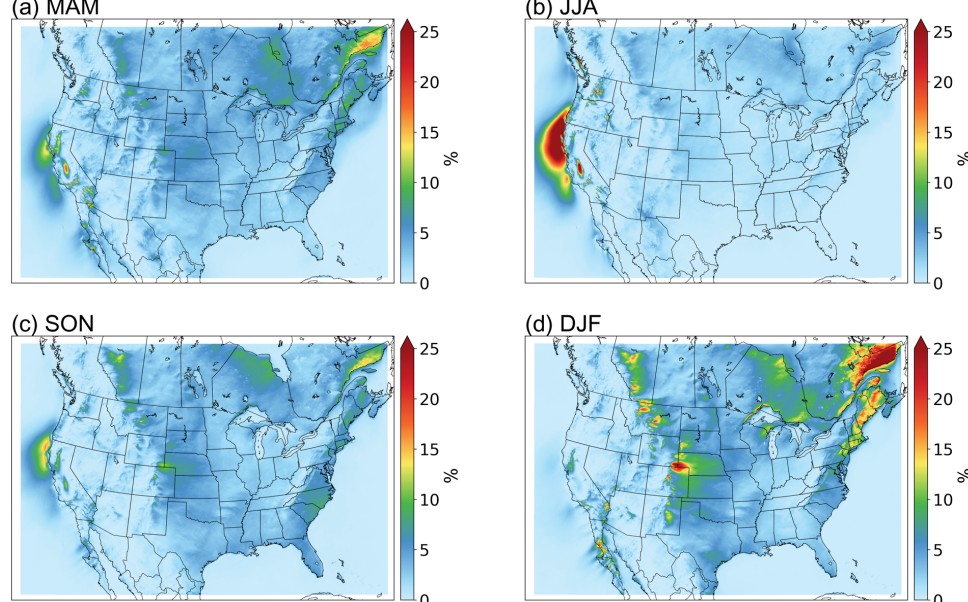


**Figure 3. Seasonal occurrence frequency of N-LLJs. Frequency shown here is calculated by counting the number of**
**occurrences of LLJs in each three-hourly time step and then dividing the total number of LLJs in each season by the number**
**of time steps in that season.**
**3.2.2 Southerly LLJs**
As to the climatology of S-LLJs in different seasons (see Figure 4), in winter, in the broad region extending from the
south Texas-west Gulf of Mexico to southern Iowa, the frequencies of S-LLJs exceed 10%. The greatest frequencies
of S-LLJs (>20%) are found along the border between northeastern Mexico and the United States. In addition, about



15% of the simulated wind profiles in south-central Texas are identified as S-LLJs. In the spring (March, April, and
May), the frequency expands significantly in >10% of areas, with clear S-LLJ distributions detected in Manitoba,
Saskatchewan, and other parts of Canada. The highest frequencies are still found in the Texas-Mexico area, where the
magnitude of these frequencies increases to over 25%. This region also extends northward to occupy most of Texas.
In winter, S-LLJs with occurrence frequencies of above 15% extend to near Colorado and Nebraska.
In summer, the area with frequencies greater than 10% no longer extends to the central Canadian prairie provinces
and Tennessee. The S-LLJs over the western Gulf of Mexico are also difficult to identify with modeled data, and their
frequency is close to 0%. In contrast, the area with frequencies exceeding 25% extends northward in summer and is
roughly divided into three parts distributed respectively in the northeast Mexico-Texas border, west-central Texas,
and the central US Great Plains (western Oklahoma and southern Kansas). The regions where more than 15% of the
wind profiles are identified as S-LLJ also expand from Colorado to near South Dakota.
In the fall, the magnitude of the frequency of S-LLJs decreases dramatically in the central US Plains and Texas. The
frequency still maintains a level greater than 15% in most areas, but with a maximum frequency of only 20% and
sporadically located in southwest Texas. The frequencies greater than 10% again expand northward and eastward in
this season, reaching Manitoba and Ontario.
There are also several S-LLJs on a smaller scale that can be seen on the seasonal S-LLJ climatology map. In spring, a
narrow region of S-LLJs with a frequency greater than 5% on the eastern side of the Appalachians extends from
Georgia through the western Atlantic to southern Nova Scotia. Over the Atlantic near eastern Maryland, the frequency
of the S-LLJ can exceed 10%. In summer, this narrow frequency belt still exists and has the same coverage, but the
magnitude of the frequency decreases and the frequency >10% is no longer visible. In winter, a region where S-LLJ
frequency is >5% extends from southwest Oregon to the west coast of British vf Canada. But in spring, S-LLJs with
frequencies >5% occur only over the ocean west of British Columbia. As for the summer, S-LLJs are almost
undetectable in this region.



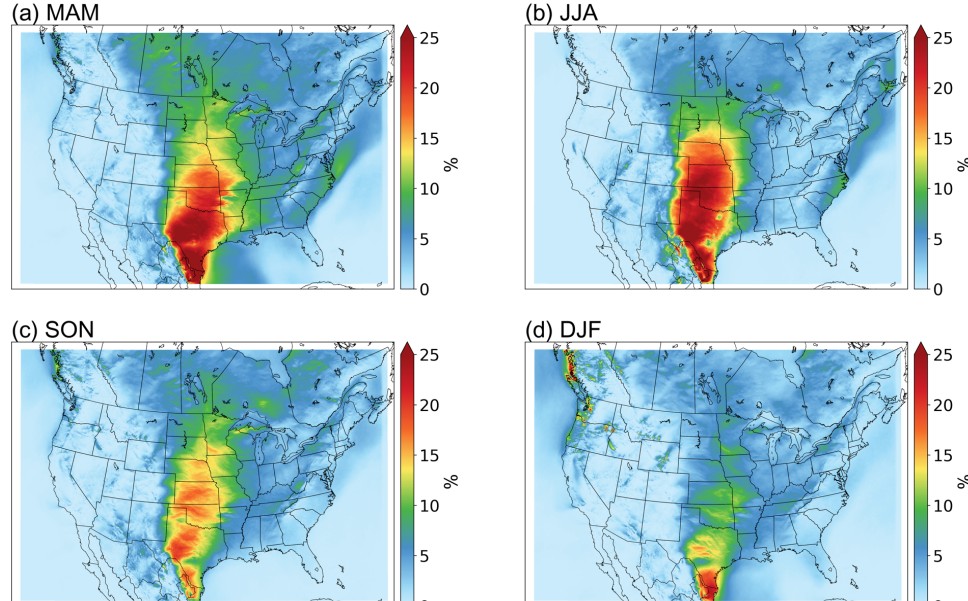

Figure 4. Seasonal frequency of S-LLJs.

To summarize, for the LLJ systems that have been investigated by many researchers, the convection-permitting WRF model performs well in observing the Great Plains S-LLJ and California coastal N-LLJ during the summer. But as to the winter LLJs that lack attention, it is essential to compare and validate the occurrence and features revealed by WRF simulation. Therefore, the ERA5 reanalysis dataset is applied in this study for capturing the LLJs in winter using the same criterion. Appendix after the text shows the results of the comparison between ERA5 and WRF simulation.

**3.3 Diurnal variations of LLJs**

To show the diurnal features of the LLJs, we selected summer and winter as the representative seasons because LLJs occur most frequently in these seasons. Below, the descriptions are divided into N-LLJs and S-LLJs.

**3.3.1 Northerly LLJs**

The California coastal N-LLJ is the most highlighted low-level jet system in this region in summer. As seen in Figure 5, it occurs throughout the day over the eastern Pacific Ocean from Oregon to the California coast. Figure 5 also shows that the California Coastal N-LLJ has diurnal characteristics: from 21 UTC, the low-level jet begins to develop, with a N-LLJ frequency of >30%, expanding until it reaches its maximum at 03 UTC – 06 UTC. Then the high-frequency



coverage of the California coastal LLJ gradually shrinks, reaching the minimum at 18 UTC and only existing off the
northwest coast of California. At the same time, the N-LLJ over the Hudson Bay Plain is also at its maximum
frequency (>5%) from 03 UTC – 06 UTC, but it rarely occurs at other time steps.

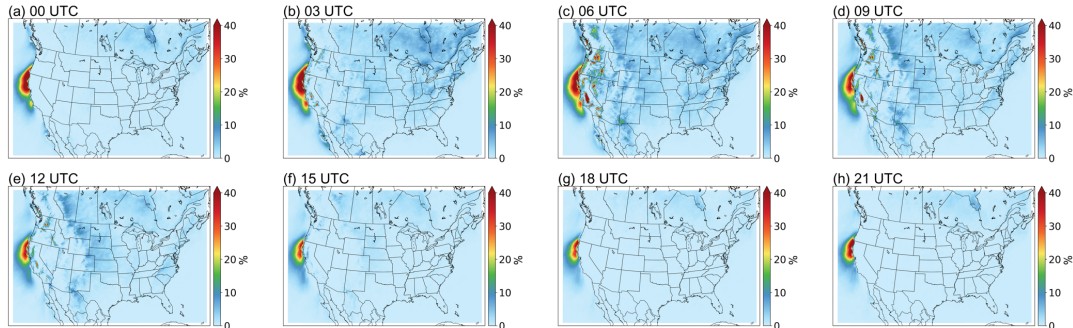


**Figure 5. Diurnal frequency of N-LLJs in the summer (JJA).**
In winter (Figure 6), three types of N-LLJs over the Hudson Bay Plain, the eastern slopes of the Quebec Labrador
Plateau, and the Appalachians display similar diurnal fluctuations. All three N-LLJs reach their highest frequency at
03 UTC and their lowest at 18 UTC. The only difference among the three types is that the smallest frequency of the
Quebec N-LLJ still endures at a level of greater than 15%, while the other two N-LLJs mostly have frequencies of
about 5%. The smallest frequency (~5%) of N-LLJs occurs downstream of the Rocky Mountains (over Alberta,
Montana, and Kansas) at 21 UTC. In the subsequent development stage, the changes in the sporadic hot spots
distributed near the eastern boundary of the Rocky Mountains are more significant. As seen in Figure 6, frequency
starts growing from 00 UTC and then peaks at 12 UTC, especially the wind maxima located in Colorado, Wyoming,
and Kansas, where the highest frequency can be >25%.

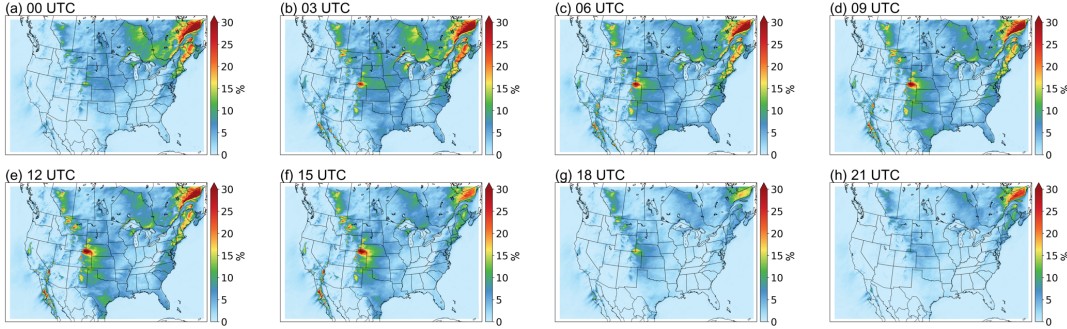


**Figure 6. Diurnal frequency of N-LLJs in winter (DJF).**





**3.3.2 Southerly LLJs**
In summer, the Great Plains S-LLJ occurs more frequently than in other seasons, and its diurnal variability is also the
strongest in this season (see Figure 7). At noon local time and in the afternoon (18 UTC – 00 UTC), almost no S-LLJs
occur over the central US (frequency <5% or about 0%). In contrast, the Great Plains LLJ begins to develop at 03
UTC, when a frequency of over 25% extends from Mexico to Kansas. It reaches maximum strength at midnight (06
UTC – 09 UTC), when the frequency reaches over 30% and the high-frequency coverage enlarges to the Dakotas, the
border of the eastern Rocky Mountains, and western Minnesota, Missouri, and Louisiana. Summer S-LLJs are also
active in southern Canada at night and in the early morning. In Saskatchewan, Manitoba, and central Ontario (03 UTC
– 12 UTC, as shown in Figure 7), S-LLJs are found with frequency >15%. In the eastern US and Atlantic, S-LLJs
occur most frequently at midnight (03 UTC – 06 UTC).

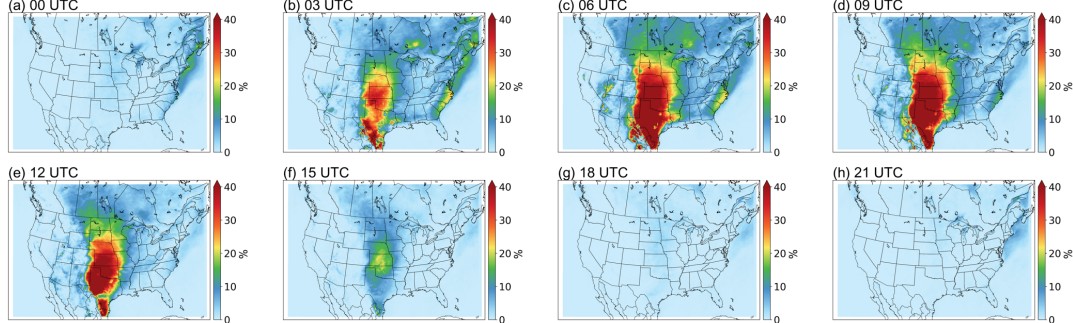


**Figure 7. Diurnal frequency of S-LLJs in summer (JJA).**
For the cold season (Figure 8), even though the Great Plains LLJ is the most inactive based on the description in
section 3.2, it still has a clear diurnal variation. Compared with the results in summer, the diurnal cycle of Great Plains
LLJ in winter is not that significant: It mainly occurs over the western Gulf of Mexico and southern Texas, with the
frequency in the afternoon (18 UTC – 21 UTC) declining to 5-10%. The S-LLJ develops from 03 UTC, gradually
generating two high-frequency (20%-25%) centers in mid- and southeastern Texas at 06 UTC – 12 UTC. As for the
S-LLJ near Vancouver Island, it is hard to see the diurnal variability: There is only a slight magnitude growth of
frequency from the afternoon (00 UTC) to the evening (06 UTC), and the coverage is almost the same.



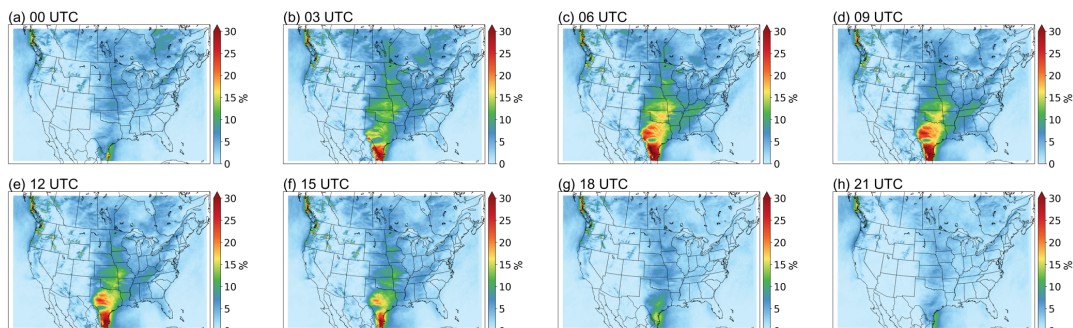


**Figure 8. Diurnal frequency of S-LLJs in winter (DJF).**

**4    Formation and evolution mechanisms of various LLJs**
Section 3's results illustrate the climatology of LLJs over North America, particularly their seasonal and diurnal
features. To explain the mechanisms, the inertial oscillation theory from Blackadar (1957) is used. Using this theory,
we start from the horizontal momentum equations and divide the actual horizontal wind $u/v$ into two components—
geostrophic wind $u_g/v_g$ and ageostrophic wind $u_a/v_a$:
$$\frac{d(u_g + u_a)}{dt} = -\frac{1}{\rho}\frac{\partial P}{\partial x} + f(v_g + v_a) \tag{1.1}$$

$$\frac{d(v_g + v_a)}{dt} = -\frac{1}{\rho}\frac{\partial P}{\partial y} - f(u_g + u_a) \tag{1.2}$$


In which $\rho$ is air density, $P$ is pressure, and $f$ is the Coriolis parameter. Assuming the horizontal pressure gradient
is fixed, the geostrophic wind is a constant as well, which means $\frac{du_g}{dt} = \frac{dv_g}{dt} = 0$:
$$\frac{du_a}{dt} = -\frac{1}{\rho}\frac{\partial P}{\partial x} + f(v_g + v_a) \tag{2.2}$$

$$\frac{dv_a}{dt} = -\frac{1}{\rho}\frac{\partial P}{\partial y} - f(u_g + u_a) \tag{2.2}$$


When the definition of geostrophic wind $u_g = -\frac{1}{\rho f}\frac{\partial P}{\partial y}$ and $v_g = \frac{1}{\rho f}\frac{\partial P}{\partial x}$ is combined, the equation (2) is:
$$\frac{du_a}{dt} = fv_a \tag{3.1}$$

$$\frac{dv_a}{dt} = -fu_a \tag{3.2}$$




If $\frac{d}{dt}$ is taken to both sides of the equations (3), then we get $\frac{d^2u_a}{dt^2} = -f^2 u_a$, and $\frac{d^2v_a}{dt^2} = -f^2 v_a$, thereby:

$$u_a = c_1 \cos(ft) + c_2 \sin(ft) \tag{4.1}$$

$$v_a = c_2 \cos(ft) - c_1 \sin(ft) \tag{4.2}$$




Therefore, according to the equations (4), the ageostrophic wind should theoretically have a circle-pattern variation
and the vector must rotate clockwise with a period of $2\pi/f$ (Blackadar, 1957; Van de Wiel et al., 2010). Under the
condition of a constant geostrophic wind—when the ageostrophic vector rotates from the opposite to the same
direction of geostrophic wind—the wind transitions from subgeostrophic to supergeostrophic. This change occurs
because of decoupling with surface friction effects, then the wind gets unbalanced.
Other theories also help explain the formation of LLJs, such as the sloping-terrain thermodynamic mechanism (Holton,
1967) and background synoptic system forcing (Uccellini et al., 1987). To understand the characteristics of the LLJs
in this study, three typical cases are analyzed: Great Plains S-LLJ, Quebec N-LLJ, and California coastal N-LLJ. The
locations for extracting data are shown in Figure 1 (solid lines and stars a, b, c).
**4.1 Great Plains S-LLJ**
As Section 3's results show, the Great Plains S-LLJ typically occurs in summer and more frequently at night. To
investigate its associated meteorological condition, this study extracts all the Great Plains S-LLJ cases occurs at the
jet core in JJA. The jet core is defined by where the mean meridional wind is the strongest on the cross-section, and it
locates at star A (shown in figure 1). The mean sea-level pressure and 800 hPa geopotential height are shown in Figure
9a and 9b, respectively. The background large-scale circulations indicate that, at all the time points when the Great
Plains S-LLJ occurs, the range of the subtropical anticyclone extends east of the Great Plains at both ground and low-
level atmosphere. A high-pressure ridge is located near the gulf coast of Mexico and Texas (Figure 9b). Thus, clearly,
the zonal pressure/geopotential gradient in the central US guides the dominant southerly winds around this region.
The cross-section in Figure 9c illustrates a strong baroclinicity and shows that the isentropic line incline moves from
east to west, as is typical for the sloping-terrain heating effect (Holton, 1967). This effect generates an upslope wind
on the east side of the slope, and the airstream gradually turns northward due to the Coriolis force, creating the
southerly LLJs. On the other hand, as can be seen in the frequency cycle in Figure 9d, at noon local time (at the





selected point-a in Figure 1), the frequency of the Great Plains LLJ is very low (close to 0%), rising to more than 40%
after 18 LST even if the radiation is not at the day's peak.

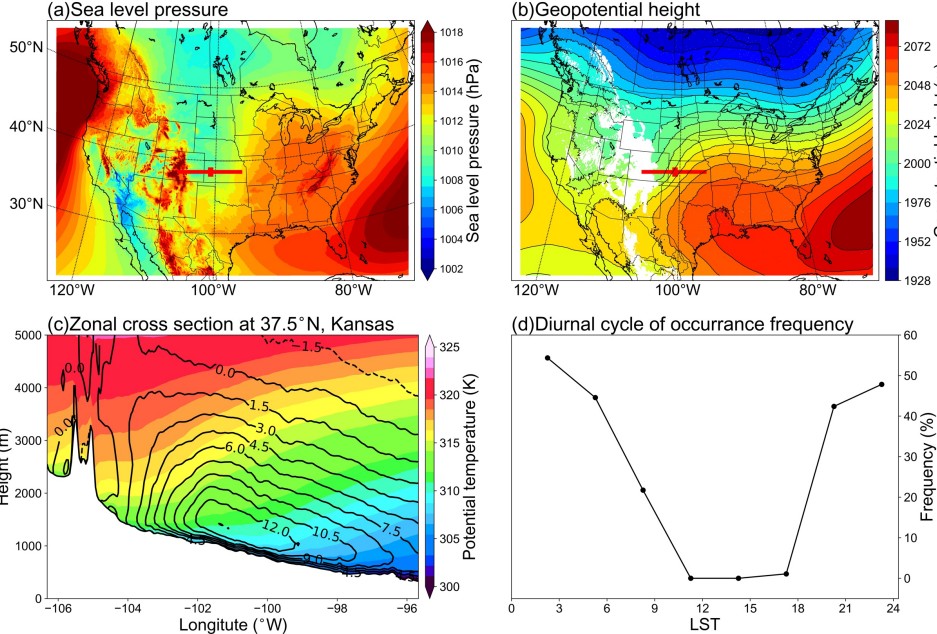


**Figure 9. Background circulations of the Great Plains S-LLJ: (a) sea-level pressure, (b) geopotential height of 800 hPa, (c) cross section including meridional winds (lines) and potential temperature (shading), and (d) diurnal cycle of frequency. The red lines and points in (a) and (b) show the position of cross-section and chosen jet core.**

To explain the nighttime enhancement of S-LLJ, we analyzed the wind vectors using inertial oscillation theory. To
show more significant diurnal variation, all the time points, including the LLJs that did not occur, were considered.
Figure 10a is the hodograph of jet-core winds at point-a near the Great Plains, and their temporal mean is computed
at 3-hourly intervals in summer. It is noted here that the "jet-core" means the position where LLJ occurs the most
frequently on the cross-section. Compared with the mean actual wind (blue arrow), the deviation at each local time
shows a clear clockwise rotation. The wind speed begins increasing after 17 LST. Nevertheless, the analysis for Figure
9 indicates the sloping heating effect, meaning that the geostrophic wind is not fixed.
Thus, to obtain the ageostrophic winds, we computed the geostrophic components by pressure gradient and subtracted
them from the actual airflow. According to the aforementioned definition of geostrophic wind, $u_g$ and $v_g$ are



calculated by the horizontal pressure gradient $\frac{\partial P}{\partial y}$ and $\frac{\partial P}{\partial x}$, respectively. By choosing four grids surrounding point-a,
we first interpolated the pressure value to the same level as the LLJ core height. Then, we adopted the central difference
equation $\frac{\Delta P}{\Delta x} = \frac{P_{i+1}-P_{i-1}}{x_{i+1}-x_{i-1}}$ or $\frac{\Delta P}{\Delta y} = \frac{P_{i+1}-P_{i-1}}{y_{i+1}-y_{i-1}}$ to obtain the pressure gradients at point-a.
Figures 10b and 10c display geostrophic wind vectors (green arrows) and ageostrophic vectors (pink) at noon and
midnight. The southerly geostrophic flows are much stronger in the afternoon (10b) than at midnight. The ageostrophic
winds flow mostly in the opposite direction, limiting the actual wind speed. At night (10c), the geostrophic wind
direction rotates clockwise from that of the afternoon as the pressure gradient changes. Considering the relative
positions of green and pink vectors at 23 LST and 01 LST, ageostrophic flow has rotated roughly 150 degrees to
enhance the geostrophic winds, thereby creating a super-geostrophic state. Although the inertial oscillation theory can
help explain some aspects of wind behavior, the real situation is more complex than initially thought. Figures 10b and
10c indicate that by 02 LST, the wind is almost entirely geostrophic with only negligible ageostrophic perturbations.
This suggests that the diurnal changes in the geostrophic wind and pressure gradient may provide a complicating
background that prevents the inertial oscillation theory from fully prevailing. While the inertial oscillation theory can
provide valuable insights, it should not be relied upon as the sole explanation for LLJs at the Great Plains. Instead, a
more comprehensive understanding of atmospheric dynamics is necessary to fully comprehend the behavior of the
wind, particularly when dealing with diurnally changing conditions. Figure 10d compares different meridional wind
components' amplitudes. The geostrophic wind contributes significantly to the southerly wind during the day, peaking
at 14 LST (green bars). The northerly ageostrophic wind (red bars) is highest during the day, indicating the strongest
negative impact from friction. The meridional ageostrophic component decreases and eventually reverses at 23 LST,
showing a process from sub- to super-geostrophic status. In summary, the thermodynamic circulation near the slopes
of the Great Plains contributes to the strong southerly airflow, while the inertial oscillation plays a critical role in
forming the nocturnal southerly LLJ.

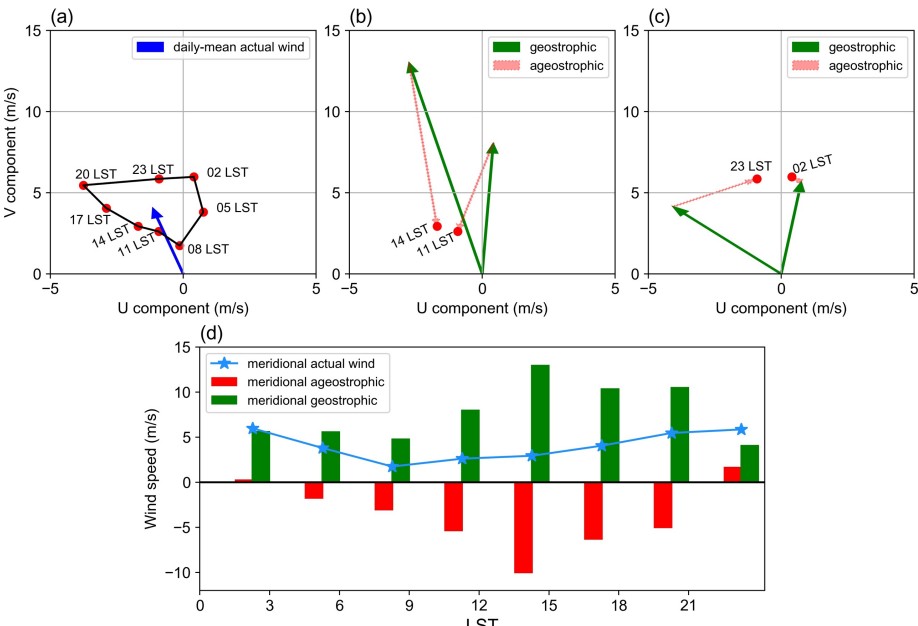


**Figure 10. (a) Hodograph of jet-core winds for the Great Plains S-LLJ every 3 hours over the whole JJA (red dots – solid line) and the daily averaged actual wind velocity (blue vector); vectors of mean jet-core geostrophic winds (solid green) and ageostrophic winds (dashed red) at (b) 11/14 LST and (c) 23/02 LST; (d) diurnal cycles of meridional components of actual (blue line), geostrophic (green bars), and ageostrophic winds (red bars).**

351

## 4.2 Quebec N-LLJ

Similarly, for the Quebec N-LLJ that is typically observed in winter, we selected all the LLJ cases at point-b (see the position in Figure 1) in DJF to generate the background circulation pattern.   The background large-scale circulations indicate that the northeastern coast of Canada lies to the west of a strong surface low-pressure system (Figure 11a), while in the lower troposphere, a ridge on the east side of Hudson Bay occupies the Labrador Plateau (Figure 11b). This combination brings the northerly momentum to the downstream eastern coast. In fact, the background circulation is consistent with the shallow baroclinic structure of Quebec N-LLJ in winter, that is, the thermal difference between warm sea and cold land. The cross-section in Figure 11c shows the thermodynamic structure of this N-LLJ: A well-defined low-level jet core is located above land and close to the coastline (approximately 63°W). With a maximum wind speed of more than 16 m s-1 and a height of about 400 m, the jet core is located above the mixed layer under the warm air covering and on the land side. Notably, the steep isentropic lines slope towards the ocean and finally sink at





the position of 60°W. In addition, the diurnal cycle of frequency (Figure 11d) shows that the diurnal signal and peak
frequency of Quebec N-LLJ are much weaker than the Great Plains S-LLJ, becoming weakest at noon and peaking at
midnight, which is consistent with the results reported in Section 3. This diurnal variation can be explained by the
baroclinicity near this region: At night in winter, the land temperature drops faster than the ocean temperature due to
radiative cooling, enhancing the land-sea contrast and thereby the thermal wind above. The gentle slope on the east of
the Labrador Plateau could generate the slope heating effect in the daytime. In this way, the related temperature
gradient from east to west offsets the land-sea thermal difference.

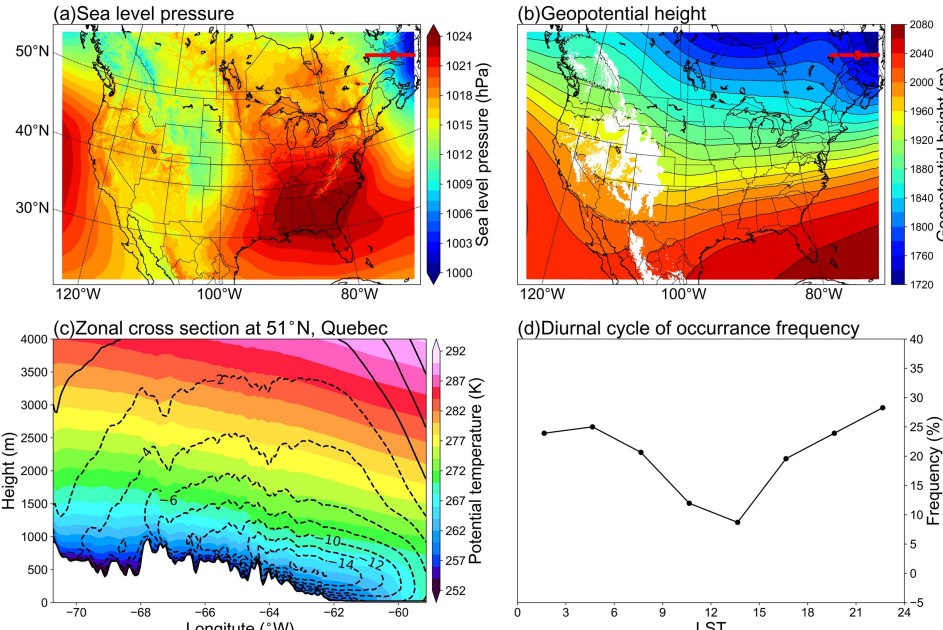


**Figure 11. Same as Figure 9 but for Quebec N-LLJs in winter (DJF).**
As for the impact of inertial oscillation on the Quebec N-LLJ, the hodograph of averaged 3-hourly winds extracted at
point-b (Figure 12a) also illustrates a clear clockwise rotation of wind deviations compared with the daily mean (blue
arrow). Figure 12b and 12c show that the geostrophic and ageostrophic wind vectors contribute to the diurnal cycle in
the afternoon and morning, respectively. Even though the direction of geostrophic wind changes significantly, the
relative angles between ageostrophic and geostrophic arrows indicate that the ageostrophic flow rotates clockwise.





The geostrophic wind is weakened by ageostrophic wind in the afternoon (Figure 12b), whereas the supergeostrophic
state is generated in the morning (Figure 12c).
Focusing only on the meridional amplitudes validates this characteristic. In Figure 12d, the blue line that represents
the mean actual meridional wind has the same diurnal trend as the frequency variation in Figure 11d. The northerly
wind is weakest in the afternoon, peaking at night and in the early morning. Similarly, the variation of meridional
geostrophic flow has a consistent phase with the actual meridional wind, which is explained by the baroclinic structure
near the Quebec coast mentioned above. The meridional ageostrophic wind in this region also promotes the formation
of N-LLJ. The ageostrophic wind drags the geostrophic component in the afternoon, before reversing to a consistent
direction with the northerly geostrophic flow at night and in the morning. This trend is also the result of decreasing
friction after sunset. Therefore, the evolution of Quebec N-LLJ derives from both inertial oscillation and land-sea
thermal contrast in winter.

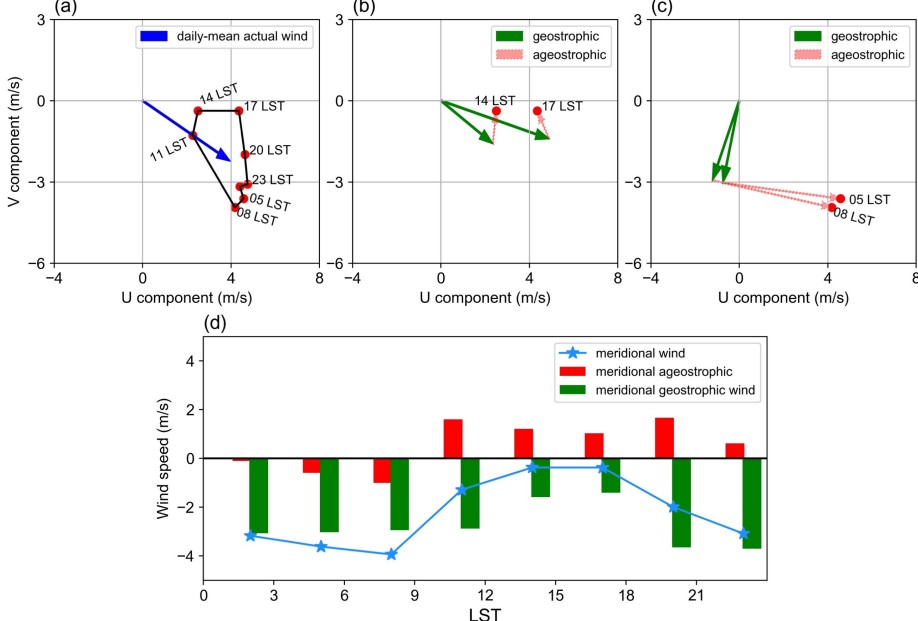


**Figure 12. Same as Figure 10 but for Quebec N-LLJs in winter (DJF).**

**4.3 California coastal N-LLJ**



The California coastal N-LLJ is similar to the one in Quebec, but it occurs more often in summer afternoons or
evenings over the ocean. Figure 13a shows that a relatively strong high-pressure system is located on the east coast of
the Pacific Ocean, trending NE-SW, although half of the structure is beyond the boundary of the domain. On the 800
hPa isobaric surface in Figure 13b, there is also an anticyclone system in the same location, whose eastern contour is
roughly parallel to the coastline, guiding the airflow to the south. Therefore, this pair is also forced by the thermal
difference between land and sea, but contrary to the LLJ in Quebec, in summer, when the California LLJ occurs
frequently, it has the characteristics of the cool sea-hot land. Figure 13b also shows that the isobars near Cape
Mendocino are relatively strong, making the ridge of high pressure extend northeastward of the Cape. This extension
is generally believed to occur due to pressure perturbation caused when northerly winds converge at this position after
being obstructed (Rahn and Parish, 2007). Regarding the cross-section structure shown in Figure 13c, the jet core is
located at steep isentropic lines above the ocean at a height of 500 m. On the coast of California, the LLJ is close to
the mountains. Compared with the Quebec LLJ, California's maximum central wind speed exceeds 20 m s-1. Based
on baroclinicity, the isentropic lines slope towards the continent and finally sink near the coastline. The core wind
speed in California's coastal LLJ may be higher than that of Quebec's LLJ because the land-sea contrast is more
significant in summer than in winter and the formed sea breeze front generates flow convergence under the blockage
caused by the west coast mountains. In contrast, the east coast of Quebec is relatively gentle, which may account for
its lower wind speed. Because California's LLJ occurs frequently (13d), the diurnal signal is weak compared, for
example, to the signal in the Great Plain S-LLJ. As well, the California signal stays at frequency of over 35%.
California's LLJ occurs most frequently at around 18 LST and starts to decline after sunset, which is generally
consistent with the coastal baroclinicity.





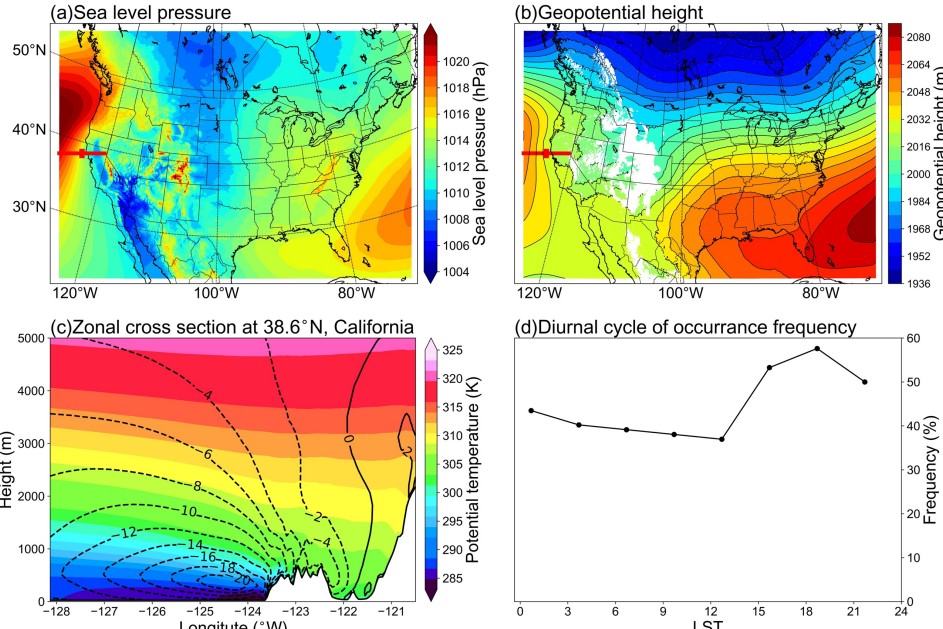


**Figure 13. Same as Figure 9 but for California Coastal S-LLJ in summer (JJA).**

The wind deviations for California's N-LLJ shown in the hodograph (Figure 14a) still have a clockwise rotation in 24
hours. However, compared with the magnitude of the daily mean jet-core wind, this diurnal cycle is not quite as
obvious as the cycle for Quebec and Great Plain LLJs, but it is similar to the frequency cycle shown in Figure 13d. In
comparison between geostrophic and ageostrophic winds (Figure.14b and 14c), during the afternoon (15 and 18 LST),
the amplitude of geostrophic wind is the largest, and the ageostrophic flow diminishes the geostrophic wind. However,
in the morning 12 hours later, the relative angle between ageotrophic and geostrophic vectors does not change,
meaning that the ageostrophic wind is still weakening the geostrophic wind and that there is no rotation of the
ageostrophic wind, as Blackadar inertial oscillation theory describes. Figure 14d helps to explain the change in
meridional winds. Looking at the magnitudes of ageostrophic winds, one can see that all are weak and southerly and
that they do not exhibit a significant diurnal signal. Furthermore, the change of geostrophic wind is highly consistent
with the trend of the actual meridional wind. Thus, the N-LLJ in California can be considered mostly as geostrophic
and the diurnal variation as being related to the change in geostrophic winds.

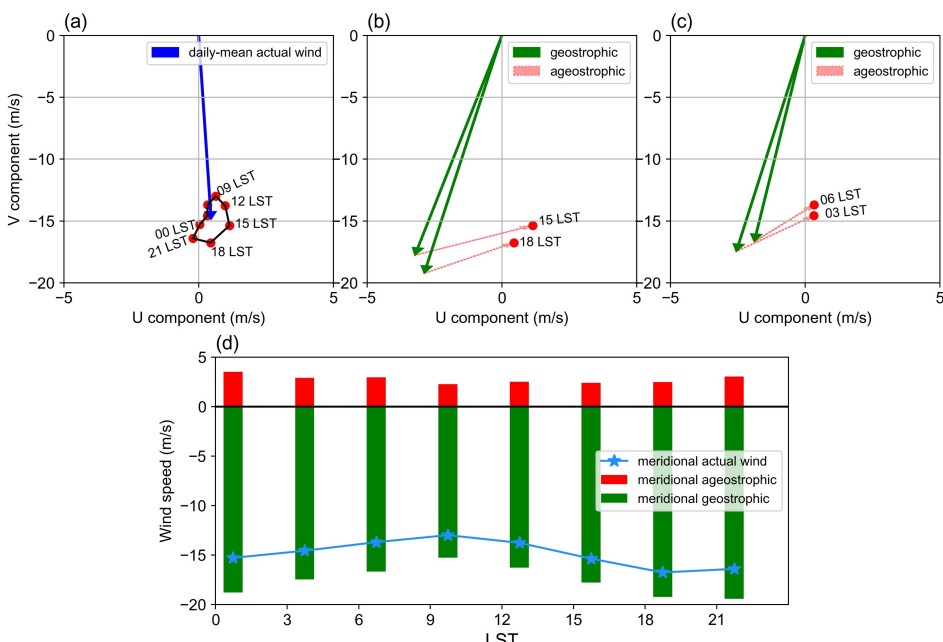

**Figure 14. Same as Figure 10 but for California Coastal S-LLJ in summer (JJA).**

## 5    Discussion and conclusion

This study applied a convection-permitting WRF model to generate the climatology of LLJs in North America. The previous research for LLJs mainly focused on observation data, which have no fine coverage in temporal or spatial resolution. The studies using in-situ observations may ignore some important features. Despite their better coverage, reanalysis datasets usually have a coarse spatial resolution, especially in the vertical direction, and can introduce large inaccuracies in the identification of LLJs. In addition, the application of general numerical modeling cannot avoid the uncertainty caused by parameterizing small-scale physical processes. In contrast, high-resolution convection-permitting climate simulations can provide more reliable and accurate descriptions of LLJs, especially for areas with complex geographic conditions or regions that lack soundings. Previous studies using high-resolution models conducted case analyses only of LLJs in a specific region (Aird et al., 2022). By expanding the target domain to the whole of North America and revealing the climatological characteristics of LLJs in different regions and scales, this paper provides an accurate reference for future research on LLJ-related processes in North America.





The convection-permitting WRF model is able to recapture some LLJs that have been previously studied, such as the
Great Plain S-LLJ and the California coastal N-LLJ in the eastern Pacific Ocean and has obtained relatively consistent
results. The results indicate that the S-LLJ in the central US Plain is the most frequent and active in warm seasons and
that three critical high-frequency centers occur in summer: the northeast Mexico-Texas border, west-central Texas,
and western Oklahoma to southern Kansas. This last result is consistent with the climatology generated by Doubler et
al. (2015) using the NARR reanalysis data, but the patterns here are more representative of the topographic features
in central and southern Texas. In addition, compared with the 40-year rawinsonde climatology in the central US by
Walters et al. (2008), our study reveals that the S-LLJ frequency range of these three centers in the central US in
summer is 25%-30%, which is slightly lower than the 35% reported in the 2008 study. However, given the
underestimated frequencies of 15%-20% in NARR climatology, there is an advantage of using high-resolution
simulations in the vertical direction.
Convection-permitting simulations can also capture the LLJs that were barely detected previously. The winter N-LLJs
over the eastern Rocky Mountains described in this paper are generally distributed over the central US from the
Dakotas to Oklahoma with a low frequency (>10%) and over several sporadic small areas with a high frequency
(>20%) along the boundary of the Rockies. The main seasonal/diurnal variations identified in this study agree with
those seen using rawinsonde data (Walters et al., 2008) and NARR reanalysis (Douber et al., 2015). But the frequency
of the LLJ occurrence over Nebraska-Kansas was underestimated in both convection-permitting simulations (~10%)
and NARR (~7%), while high-frequency hot spots from Alberta to Colorado were not detected in either of the above-
mentioned studies, probably because measurements are lacking in these regions. The high-resolution simulation also
detected LLJs on which researchers have hardly focused: N-LLJs near the eastern Quebec coast and in the
Appalachians Mountains, as well as an S-LLJ over the British Columbia coast. In the work of Douber et al. (2015),
these LLJs were shown in the climatology patterns, but the 4-km WRF simulation offered more detailed descriptions
of their locations. For example, this study found that the Appalachian N-LLJ extends from Georgia to the northwestern
Atlantic, especially on summer nights (03 UTC – 06 UTC), while NARR only captured LLJ occurrences over the
middle coast of the Atlantic. The maximum frequency (7-10%) detected in the NARR study is also less than what is
illustrated here. As for the Quebec N-LLJ, the 4-km WRF revealed that it mostly occurs onshore near the coast with
a frequency of over 25% in winter, but NARR only provided a coarse occurrence distribution over northeastern Canada.



To investigate the significance of LLJs in different regions, Figures 15 and 16 demonstrate the impact of the Great
Plains S-LLJ and Quebec N-LLJ, respectively, on downstream extreme precipitation during their active seasons.
Figure 15a illustrates the 90th percentile of summer precipitation in the central United States, indicating that 90% of
the precipitation in most areas falls within the range of 1.0-2.0 mm/hour. However, Figure 15b shows the ratio of
strong events related to LLJs (counted if the precipitation is > 90th percentile when a LLJ occurs) to all strong events,
with the red outline on the map indicating the approximate location of the low-level jet stream. It is evident that in the
lower reaches of the S-LLJ in the Great Plain, particularly in the north-central United States, nearly 50% of the heavy
precipitation events are associated with the flourishing low-level jet stream. Furthermore, Figure 15c displays the
average precipitation of all LLJ-related strong events. Compared with Figure 15a, some areas of Nebraska and
Minnesota experience rainfall of up to 6mm/hour. These findings highlight the significant role played by LLJ in
modulating summer precipitation. Similarly, for the Quebec N-LLJ in winter (Fig. 16), it contributes more than 25%
of the strong events of precipitation in the Gulf of St. Lawrence during winter (Fig. 16b). Figure 16c further reveals
that, in comparison to the 90th percentile rainfall, the extreme precipitation from Quebec to Maine is approximately
1mm/hr higher. Particularly during the cold season when a substantial portion of precipitation is snow, the N-LLJs
can also be seen as the factors of snowstorms in this region. In summary, research on the importance of LLJs includes
not only the field of extreme precipitation, but also local wind energy production, air pollution dispersion, wildfires,
etc. (Jain & Flannigan 2021, Lin et al. 2022, Weide Luiz & Fiedler 2022). There is no doubt that the high-resolution
regional climate model presented in this paper provides ample coverage and details about LLJs in North America, to
support analysis in these fields, particularly at the national level. With a grid spacing as small as 4 km, researchers can
even employ the wind profiles from model output to investigate small-scale areas, such as wind farms or wildfire
ignition sites.

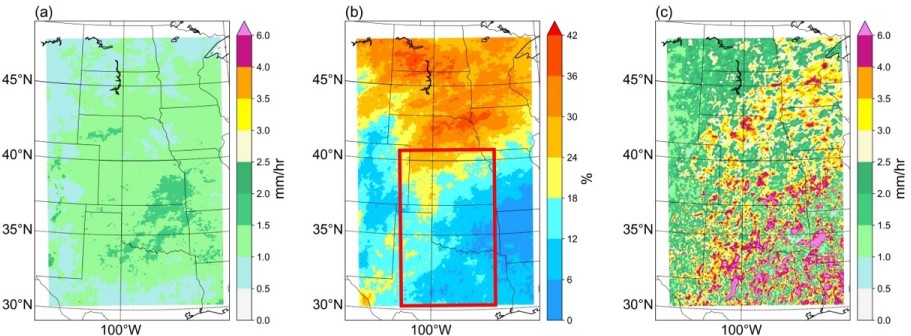

**Figure 15. (a) 90th percentile of summer precipitation rate over Central US; (b) The ratio of LLJ-related strong rainfall events to all strong events, the red outline represents the location of Great Plain S-LLJ; (c) Averaged precipitation rate of LLJ-related strong events.**

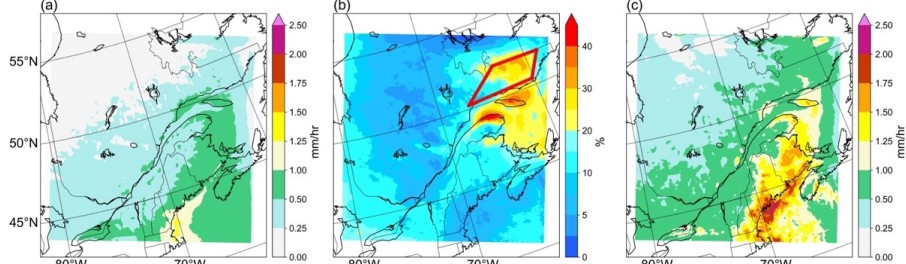

**Figure 16. (a) 90th percentile of winter precipitation rate over Southeastern Canada; (b) The ratio of LLJ-related strong rainfall events to all strong events, the red outline represents the location of Quebec N-LLJ; (c) Averaged precipitation rate of LLJ-related strong events.**

Based on the inertial oscillation theory (Blackadar, 1957) and the baroclinic theory near complex terrain (Holton, 1967), this paper also analyzed the background and formation mechanisms of three LLJs: the Great Plain S-LLJ, Quebec N-LLJ, and California coastal N-LLJ. Generally, all these LLJs are impacted by the thermodynamic circulations generated near their topography. The Great Plain S-LLJ is affected by slope heating, and the LLJs over Quebec and California are associated with the sea-land contrast. When the geostrophic and ageostrophic components of the LLJs are compared, results show that the inertial oscillation better explains the night enhancement of the Great Plains S-LLJ and that the diurnal feature of the Quebec N-LLJ is influenced by the combination of the Holton and Blackadar theories. As for the California coastal N-LLJ, no supergeostrophic state is found, making coastal baroclinicity variation a dominant factor for this LLJ's evolution the geostrophic wind changes.



The LLJs climatology introduced in this research adds to the existing knowledge of characteristics of the low-level
wind maxima in North America, thus helping researchers obtain more reliable references about LLJs in this domain.
Meanwhile, with the high-resolution features, it can provide more robust explanations for other interdisciplinary fields.
The research also advances knowledge about the formation of three dominant LLJs. Although the 13-year simulation
is likely too short to provide an ideal long-term climatic analysis, it is a less expensive option for finer numerical
modeling in large domains. But it is also believed that with the advancement of technology, there will be longer high-
resolution simulations in the future. Future work will address the features and formation mechanisms of the small-
scale low-level wind maxima that have yet to be investigated.



**Acknowledgments**
All authors thank the support of the Global Water Futures Program by the Canada First Research Excellence and the
NSERC Discovery Grant.

**Data Availability Statement**
The WRF simulation over CONUS can be accessed at Research Data Archive of NCAR
https://rda.ucar.edu/datasets/ds612.0/.

**Author contribution**
Xiao Ma: Conceptualization; data curation; formal analysis; investigation; methodology; visualization; writing-
original draft.
Yanping Li: Conceptualization; funding acquisition; investigation; methodology; project administration; supervision;
validation; writing-review and editing.
Zhenhua Li: Data curation; methodology; validation; visualization; writing-review and editing.
Fei Huo: Data curation; methodology; validation; visualization; writing-review and editing.

**Competing interests**
All authors disclosed no relevant relationships.



**Appendix**

**Winter LLJs captured by ERA5 Dataset**

The convection-permitting WRF simulation exhibited excellent performance in investigating well-known LLJ systems, such as the California coastal N-LLJ and the Great Plains S-LLJ. Moreover, this appendix validates WRF-simulated significant winter jet systems over North America using the ERA5 reanalysis dataset. ERA5 is a global atmospheric reanalysis dataset produced by the European Centre for Medium-Range Weather Forecasts (ECMWF). It provides hourly data on a horizontal grid space of approximately 31 km, and the time range covers from 1979 till the present. ERA5 data is widely used in climate research, weather forecasting, and various applications that require high-quality atmospheric data.

The validation period is the same as the WRF simulation (2000-2013). From the Figure A1 below, it is evident that during winter, a greater number of significant N-LLJ systems in the North American continent are mostly concentrated in eastern Canada. In most parts of Newfoundland and southeastern Quebec, the occurrence frequency of N-LLJs exceeds 15%, and the maximum can even surpass 25%. However, in the WRF simulation (Figure 3d), the model can only capture N-LLJs on the north bank of the St. Lawrence River due to the northern boundary of the study domain overlapping with the Quebec border. In comparison, the WRF-simulated frequency of N-LLJs in southeastern Quebec essentially exceeds 25%, overestimated by about 5% compared to the ERA5 reanalysis. Additionally, it is worth noting that the N-LLJs along the downstream of Rockies are also identified in the ERA5 dataset. The areas where the frequency exceeds 5% are mainly distributed from Alberta to northern Texas, consistent with the findings in Section 3.2.1. Moreover, the high-value center (>10%) is located in central Kansas. In terms of the differences between the two datasets, the results of the WRF simulation match more geographical features and reveal scattered high-value spots (>15%) in some regions with special terrains (see Figure 3d). Furthermore, the winter Great Plains S-LLJs in ERA5 reanalysis exhibit similar features, with frequencies ranging from around 15% to 20% in southern Texas. In summary, the WRF model can accurately capture the features of winter LLJ systems, which are validated by the ERA5 reanalysis dataset over northern America. Even though the frequency of LLJs occurrence is overestimated, the convection-permitting WRF simulation can provide detailed descriptions of LLJs near complex terrains.



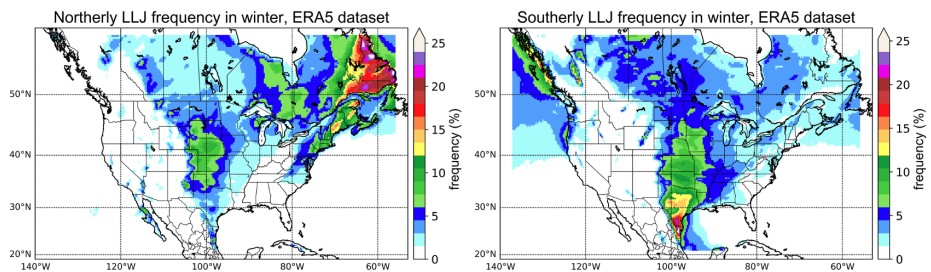


**Figure A1. Winter occurrence frequency of N-LLJs (left) and S-LLJs (right).**


**Data Availability Statement**

The ERA5 dataset is available on the Copernicus Climate Change Service Information website.
https://cds.climate.copernicus.eu/#!/home



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
