# Peer review of "Investigation of the climatology of low-level jets over North"

_EGUsphere, 2023_

## Author Response (AR1)

Dear editor,

Thank you for your patient response to my revision. I am very appreciated to you and the reviewers for the helpful comments and further clarifications provided!

Here are responses to the reviewers' comments:

**Reviewer #1:**

Major Comments:
Title and throughout the manuscript: Since only 13 years of data are used for the analysis, I think the authors should be very careful calling the work a climatology (which you also mention, line 509-510). The standard normal is to use 30 years for a climatology. All places in the text where climate/climatology/climatological is mentioned should be carefully checked and changed appropriately.

While a 13-year simulation may not fully suffice for comprehensive climatological analysis, it represents a pragmatic approach given the substantial computational demands of convection-permitting simulations across a broad study area. The challenge of executing high-resolution simulations over several decades renders 13 years a balanced compromise. Acknowledging this limitation in the discussion, we will extend our research to longer-term, large-domain regional simulations as advancements in technology enhance computational feasibility in the future.

Why did you use ERA-Interim? ERA5, the successor of ERA-Interim, is state-of-the-art and has been around for many years now. It also has higher spatial and temporal resolution than ERA-Interim, which is important for your study. It's not clear why you don't perform the downscaling based on ERA5. The choice of using ERA-Interim instead of ERA5 needs strong motivation.

Yes, indeed, ERA5 is a popular and powerful input data currently. In this study, we used the 4-km WRF simulation dataset generated by Liu et al. (2017), which used ERA-Interim to drive the model while ERA5 was unavailable then. There are many studies based on it. ERA-Interim has been accumulating a wealth of application cases and experiences. Of course, the simulation based on ERA5 will be the focus of our future work, but here, according to the results part, our ERA-Interim-based work is robust.

The analysis of northerly and southerly LLJs (N-LLJs and S-LLJs) is very

interesting and important. However, easterly LLJs (E-LLJ) and westerly LLJs (W-LLJ) are completely neglected. The bins should be +/-45° around the four cardinal wind direction. Including all wind directions in the analysis is needed to strengthen this manuscript and is crucial for publication. Further comments on this topic are given below.

This classification criterion, based on the study of Walter et al. (2008) and Doubler et al. (2015), is commonly used in relevant research , and it considers the meridional LLJ for heat and water vapor transport . In this paper, we  compared  our results with previous studies to examine if the LLJ capture is robust. Therefore, the results are more comparable when using the same standard in this study. For the details of the standard: southerly LLJs (S-LLJs), the jet-core wind direction is between 113° and 247°; for northerly LLJs (N-LLJs), the jet-core direction is between 293° and 67°. This means most (about three-fourths) of the wind directions have been considered in our study by this criterion.

L468-482 and Figures 15 and 16: This comes across to me as a rather different topic from the rest of the work. I strongly suggest to cut it from this study. It is interesting but should be a paper on its own.

In fact, the purpose of these two figures is to supplement the valuable context and clarification that could be beneficial for readers in understanding that Quebec N-LLJs are also important to understand local winter precipitation and climate, offering the reference to meteorologists in this region. Though less influential and less studied, the role of LLJs near Quebec is an interesting topic. For example, in the discussion part of this manuscript, Quebec N-LLJ is shown to contribute more than 25% of the strong events of winter precipitation in the Gulf of St. Lawrence. This can be a good factor to predict the snowstorm in this region. Therefore, I keep the results. On the other hand, this analysis also offers new ideas that we may extend the research in the future.

Minor Comments:

1. Title: I think a title that better describes the uniqueness of the work is preferable. A suggestion is Investigation of the occurrence of low-level jets in different wind directions over North America

   We agree with your view on the uniqueness of the title. The highlight of this paper is the application of a 4km convection-permitting model, so we have changed the title to "Investigation of the climatology of low-level jets over North America in a convection-permitting WRF simulation"

2. L26-27: It sounds like an LLJ is always present in the lower atmosphere, which is maybe not always the case? Please rephrase if LLJs only appear

sometimes or even most of the time.

Thanks for your correction. This sentence has been revised to "A low-level jet (LLJ) is described as the fast-moving air ribbon located in the lower atmosphere most of the time (Bonner, 1968; Rife et al., 2010)."

3.  L29: A lot of research has also been performed to study the offshore LLJ. This should also be mentioned, with appropriate references.

    References added: "... Besides, not only these in-land LLJs are investigated, but also offshore coastal LLJs such as the California LLJs (Parish, 2000) and North African Coastal LLJ (Soares et al., 2018) ..." Thanks!

4.  L31: Reference needed for "but its width can reach several hundred kilometers"

    Sorry, it was a typo. The description should be "but its length can reach several hundred kilometers", rather than "width".

5.  L33-35: Please extend and briefly explain how LLJs affect wind power, air pollution, and urban heat islands.

    More detail added:
    "Meanwhile, researchers have long been interested in investigating their features, because LLJs also affect various processes such as wind power development, air pollution transportation, and urban heat islands: the wind turbines would be influenced by positive wind shear and downward entrainment from the LLJs above them, assisting in extracting energy from the strong wind belt inside LLJs (Gadde and Stevens 2021; Ma et al., 2022). LLJ-related horizontal transportation is beneficial to pollutant removal (Sullivan et al. 2017). The LLJs can enhance the turbulent mixing in the boundary layer thereby decreasing the atmospheric stability, helping pollution diffusion, and weakening urban heat island intensity (Hu et al., 2013)."

6.  L35: A great part of the LLJ research has focused on only the lowest 300, 500, or 1,000 m of the atmosphere. It should be mentioned already here in the Introduction that wind profiles all the way up to 3,000 m are assessed for LLJs in this study.

    In fact, according to what you mentioned, the research on LLJs is concentrated below 1 km because the core of most LLJs is at this location. But the standard definition of 3 km here means that I need to find the two minimums of the wind profile within a height of 3km to identify a "nose". In other words, the minimum wind speed above the jet core needs to be found within a height of 3km. Actually, the core heights of the low-altitude jets

found in this study are all below 1km, which is consistent with other studies.

7. L40-46: This part feels a bit out of place to me, and should be removed as you are not using any observational data in your study.

Thanks. In my opinion, the application of observation data described here mainly emphasizes the advantages of model data in space and time, so I think mentioning these in the introduction can make a good comparison before and after, and at the same time form a progressive relationship with the following text.

8. L55: Also lidar measurements, which can be used to study LLJ in the lowest approximately 300 m of the atmosphere, should be mentioned here.

Yeah, I have mentioned lidar as you suggested:
"...Although observation platforms such as radar, PECAN, or lidar which investigate the atmosphere as low as 300 m, can compensate to some extent for this lack of observational data. as well as lidar that investigates the atmosphere as low as 300 m, these approaches are still limited by the spatial coverage of their measurement platforms (Smith et al., 2019)."

9. L57-59: "Reanalysis data [...] perform more extensive measurements" Reanalyses don't perform measurements, so I'm not quite sure what you mean here. Please reconsider the formulation of this sentence.

Thanks for pointing this out. I have revised the text to:
"Reanalysis data have relatively better spatial and temporal coverage than rawinsonde measurements, incorporate observations into the preliminary model simulations, provide more comprehensive variables through assimilation, and contain broader domains."

10. L61: "previously unknown jets" Please add information about which these unknown jets are.

Sure, I have listed the unknown LLJs in the sentence, they are "Tarim nocturnal LLJ in northwest China, Ethiopia nocturnal LLJ, and Namibia–Angola nocturnal LLJ."

11. L77: Add a reference for the "shows promise" statement

Actually, the following reference about Du and Chen (2019) is an example of using the convection-permitting model, they highlighted the coastal terrain. Maybe "shows promise" is not an accurate description here, so I replaced "promise" with "ability".

12. L80-81: References to studies showing that models with higher resolution show more precise results regarding LLJs should be added here.

This sentence is actually the extension of the previous reference about Du and Chen (2019), so I did not add the literature here.

13. L82-96: This description of the formation mechanisms should be presented earlier in the Introduction (perhaps after L35?). Further, coastal LLJs, cold front LLJs, and typical formation processes of offshore LLJs should be explained here.

Because the literature review on the LLJ mechanism is very long in the introduction, I think the entire article will look a bit confusing if it is placed earlier. My logic is still to first describe the observations and simulations of LLJ, and then describe the mechanism. This order will also be consistent with the order in the results section below. Meanwhile, the purpose of this paper is not a pure literature review of LLJ classification and formation, so I do not tend to explain every aspect in the introduction part. Thank you for your advice.

14. L107: Did you use any nesting in your downscaling? It should be stated here as well.

Sure, the simulation did not use nesting, I have revised the sentence in section 2. Thanks!

15. L109: Please change throughout the entire manuscript and refer to your work as "downscaling" of ERA-Interim rather than a "simulation".

Thanks, I think the dynamic downscaling of ERA-Interim data can be described as "simulation", because the whole procedure was just inputting coarse large-scale ERA-Interim and conducting the simulation on the smaller grid space. It utilized the physical processes of numerical weather prediction. So "simulation" word in this paper is accurate.

16. L110-112: "five layers under 500-m height and nine layers under 1 km are outputted above ground level, which means the WRF has the good ability to capture the LLJs occurring in the boundary layer". No! This statement is not correct, because a minimum of three levels are required to identify an LLJ and the height of the boundary layer can in some circumstances be very low (even below 100 m). On top of that LLJs can also be very narrow in height, in some cases the core is only 50-100 m in its vertical extent.

Thanks, here is just for a comparison, because compared with other modeling data, this density of vertical layer can really capture more LLJs in the lower atmosphere as you said. But just relatively more, so maybe in the absolute standard, 4km WRF is not good, but maybe it is "better ability than GCMs/RCMs". So I revised the description here

17. L113: What was the vertical resolution of ERA-Interim used as input data for the WRF downscaling? Please add this information.

Sure, in this paragraph I have added "…, the vertical layer depth of inputted ERA-Interim data under 5 km is about 0.3-1.4 km (Hoffmann & Spang, 2022)."

18. L134: At which height level is the wind direction used for classifying the LLJ as S-LLJ or N-LLJ? Or is it the wind direction at the core of the LLJ? Please clarify.

Yes, the LLJ is classified by the jet core wind direction, I have clarified this in the manuscript now, thank you!

L142-157: This analysis and Fig. 2 is not relevant and the connection to LLJs has not been motivated. Following the major comment above that the study should focus on LLJs from the four wind directions (N, E, S, W), maps showing the relative occurrence of those wind directions for the different seasons should be included instead.

Even though Fig. 2 is not really relevant to the results of LLJs, it is able to be shown as proof of the ability of WRF simulation, by illustrating the large-scale circulation over North America. On the other hand, the past studies of LLJs have used the same criterion in this paper, which only focuses on the Northerly and Southerly LLJs. Under the same standard, it is easier to have the comparison of models' performance. So your suggestion of four directions seems not necessary, but I am grateful for your suggestion!

19. Fig 3.: I suggest a 4x4 panels figure showing the occurrence of LLJs in different wind directions as the rows (N, E, S, W) and different seasons as the columns (DJF, MAM, JJA, SON). This would simplify comparisons between the occurrence in different wind directions and in different seasons. For Fig.3, white color could be used for the lowest percentages (0-1%) and a discretized colorbar (as in Fig. A1 in the Appendix) would increase the readability of the plots. Further, a 4x4 panel figure showing the average LLJ core height (fore sites with at least 5 or 10% relative occurrence) in different seasons and in different wind directions would be valuable and could strengthen the discussion.

In response to the last comment, I have already clarified the reason why I chose only two directions of LLJ-core wind. As to using the discretized color bar, in this paper, we want to highlight the good performance of 4-km WRF in capturing the LLJs over complex terrains. Therefore, compared with the discretized color bar the sequential color bar has a better presentation for small-scale details. For example, in Figure 3, we can see the shape of mountains in the LLJ frequency colors, thus the result would cause readers to consider if the LLJ is related to the orography.

20. L210: "British vf Canada", what is "vf"?

    Sorry, it is a typo, the original word should be "British Columbia of Canada". Thanks for correction.

21. L215-219: This validation doesn't hold. If ERA5 is trusted as the best description of LLJs, why don't you use ERA5 in the first place? Validation of the WRF downscaling has to be against observations (e.g., radio soundings), potentially showing superior performance of the WRF downscaling as compared to ERA5 or ERA-Interim. Following this remark, the Appendix should be removed.

    Thanks, the purpose of the appendix is to prove the performance of WRF simulation in capturing basic LLJs, rather than simply comparing the advantage of the convection-permitting model. Indeed, ERA5 is a much better dataset than others, the reason why we don't use ERA5 in the first place is actually the climate simulation can have future results, and the future change in LLJs is also under consideration. So we applied the 4km WRF model, and examined if WRF can capture basic LLJs in the appendix, and then in the future work we will discuss the impact of climate change. This is the logic flow of our project.

22. L222: In summer, yes, but it is not true that S-LLJs appear most frequently in winter, so please rephrase this sentence.

    Sure, I have rephrased the sentence as:
    "To show the diurnal features of the LLJs, we selected summer and winter as the representative seasons because S-LLJs and N-LLJs occur most frequently in these seasons, respectively. Below, the descriptions are divided into N-LLJs and S-LLJs. "

23. L226: Please also state in the text what 21 UTC is in the local time zone for California.

Thanks for the reminder, 21 UTC is 1 pm in the local time zone for California, I have added the statement in the manuscript as:
"... from 21 UTC (1 pm LST in California), the low-level jet begins to develop, with a N-LLJ frequency of >30%..."

24. L229-230: It is not possible for me to spot the Hudson Bay N-LLJ. Remove this sentence as you are not describing other LLJs that appear only approximately 5% of the time.

Yes, I removed this description about an insignificant LLJ in the revised paper, thanks.

25. L233: I think Hudson Bay Lowlands is a more common name than Hudson Bay Plain, so please change.

Sure, the "hudson bay plain" has been replaced with "hudson bay lowlands".

26. L235: Again, please state what 18 UTC is in local time.

Yes, because Hudson Bay Lowlands, Quebec Labrador Plateau, and the Appalachians are all in the Eastern Time Zone, so the local time is 5 hours behind UTC. I have stated the correct local time in the paragraph.

27. L258: Maybe change "significant" to "pronounced" instead? Unless you performed some significance testing?

Sure, the word "significant" has been replaced as advice, thanks.

28. L266-293: This part should be moved to form a new Theory section, placed between Sects. 1 and 2.

Thank you very much for this valuable suggestion. After consideration, we have decided to maintain the current structure. I think the related mechanism theory in its current location can give a more intuitive explanation for the subsequent results part. If I put the theory part after Section 1, it may be hard for readers to refer to it when they scan to Section 4. Furthermore, I am also concerned that reorganizing the sections might disrupt this balance among sections, resulting in previous sections being lengthy or information-dense, potentially affecting readability. But I am very grateful for your concern here!

29. L297: Please refer to the figures where this can be seen.

Sure, Figure 7 should be referred to here, I have added relevant words,

thanks.

30. Figures 9, 11, and 13, panels a and b: Plot anomalies in panels a and b instead of the average fields, to show how much the pressure or geopotential height differs compared to normal in cases when an LLJ is present.

    The anomaly field will indeed provide more significant patterns than the climate mean field, but here I still hope to retain the climate mean field, because then I can refer to more large-scale climate systems for helping to analyze the results. In addition to discussing the relationship between LLJs and large-scale circulation, we can continue to use these figures as performance proof of WRF simulation. Thanks for your suggestion here.

31. Figures 9, 11, and 13, panel c: Please also show the location of the selected site along the x-axis.

    I have used one vertical line in subplot c to represent the zonal location of the selected jet core.

32. Figures 9, 11, and 13, panel d: Add confidence interval to show if the diurnal cycle really is significant.

    I have used the gray shading to show the confidence interval of the diurnal cycle plot.

33. Figures 9, 11, and 13: In addition to panel d, which is very interesting, also include plots showing the diurnal cycle of core speed and core height. Also, in panel d, let the line extend over the left and right edges to show the changes there (i.e., in Fig 9d, 23 LST and 2 LST should be "connected" with a line).

    Yes, now in subplot d of Figures 9, 11, and 13, the lines are connected by extension.

34. Fig 9: Please add in the caption that this is for the JJA season.

    Thanks, I have added the season information in the caption.

35. L318-319: Why not use the actual core height at every given time step? It might vary drastically from the most frequent core height.

    Thank you for your question. In the manuscript, the position of the "jet-core" I explained refers to the position in the horizontal direction where LLJ occurs most frequently, which is the point in Figures 9a and 9b. And the jet-core height is indeed determined at each time step. So it is calculated as you said

in the question. But obviously, something is confusing in my manuscript, so I've added "horizontal" to this sentence to make it clear.

36. L326: Please clarify by adding "where i is the index of the grid point at point-a".

Sure, thanks for your clarification, I have finished for this!

37. Figures 10, 12, and 14, panels abc: Please scale so that 5 m/s on the x-axis has the same size as 5 m/s on the y-axis.

I appreciate your suggestion. However, it's important to note that the meridional components of the Great Plain S-LLJ and California coastal N-LLJ significantly outpace their zonal counterparts. Matching the y-axis ticks to the x-axis would elongate the geostrophic wind vectors in these scenarios, aligning them almost parallel to the y-axis. At the selected local time, the ageostrophic wind vectors would appear negligible next to the geostrophic ones, obscuring the intended results. This rationale underpins my decision to maintain the current y-axis tick scaling. Thank you for your input!

38. Figures 10, 12 and 14: Please adjust the colours to account for people with color vision deficiency, red and green is unfortunately not the best combination.

Sure, I have used red and royal blue to make the contrast of different winds in this kind of figures. Thanks!

39. L353 (and elsewhere): Please note the N-LLJ is often used as an abbreviation for nocturnal LLJ. I suggest switching to write LLJ(N) (and similar for the other wind directions) which would simplify for the reader.

Thanks for the reminder. In some papers, the writers used N-LLJ to represent the nocturnal LLJ, but it can also be the abbreviation of Northerly LLJ, for example, in the paper named "A Long-Term Climatology of Southerly and Northerly Low-Level Jets for the Central United States" (Walters et al., 2008). I think if I explain clearly at the beginning, it would be okay.

40. L359: Add "relatively" before "warm sea".

Sure, thanks! I have finished for this!

41. General for Sect. 4.1-4.3: Please also comment on the stability of the atmosphere when LLJs are occurring (panels c in Figures 9, 11, and 13)

Sure, I have added the description in sections 4.2 and 4.3 to discuss the atmospheric stability by isentropic lines. And try to explain the different wind speeds of three cases by this stability theory.

42. Figures 11-14: Please add full captions for all these figures (even if it is repetitive it is easier for the reader than having to scroll back and forth).

    I have finished revisions you suggested here, thanks.

43. L392: How is it similar? It's not clear to me.

    Well, here the word "similar" means California coastal N-LLJ is also located upon the contrast between sea and land, even though it is offshore but Quebec N-LLJ is onshore mostly. But they may have a "similar" mechanism of formation, this is what I want to state in this sentence.

44. L403: "Compared with the Quebec LLJ, California's maximum central wind speed exceeds 20 m s-1". This sentence should be rephrased as it is not a comparison.

    I have rephrased the sentence into "The maximum central wind speed of California coastal LLJ exceeds 20 m s-1, whereas Quebec N-LLJ's max core wind is only about 14 m s-1."

45. L408: The fact that California's LLJ occurs frequently cannot be the reason for that the diurnal signal is weak.

    Here I just wanted to present that the California N-LLJ occurs much frequently at each time step, and then the diurnal cycle did not look obvious in Figure 13d. There is no causation between these two descriptions. But this sentence is weird and confusing, I have modified it to "California's LLJ occurs frequently at each time step, its diurnal signal is weaker compared, for example, to the signal in the Great Plain S-LLJ". Thanks for correction.

46. L433: The model level output from ERA5 actually has a better vertical resolution than the WRF output you are using, so this sentence should be expressed in a more careful way.

    I understand, so I removed the phrase "...especially in the vertical direction, ..." to make the whole sentence more accurate. Thanks!

47. L435-436: It is not clear how using a convection-permitting model actually gives better resolved LLJs. Since LLJs tend to appear in stable conditions (where the convection-permitting features shouldn't be important), this

statement has to be stronger motivated.

Thanks for your suggestion, so "more accurate" in the original manuscript may not be that motivated, I used "relatively more comprehensive" to replace this phrase. At least considering the spatial resolution and temporal density, the convection-permitting model can really give "more comprehensive" results.

48. L447-449: The numbers presented here are interesting, but since you did not study the exact same time period it is difficult to draw any conclusions based on this. A comment about this should be added here.

Thank you for your comments. I admit that the data presented in this study are not from the same period, which is indeed important for comparing results. However, for climatological research, even data over different periods can still provide valuable insights and references. This is because climatology focuses on long-term trends and patterns that often transcend specific time frames. Therefore, despite the different periods, a comparison of these data still reveals some key features.
During the comparison, I have tried to keep the observation time consistent, because the radiosonde data only has two observations a day, and the occurrence frequency I mentioned here is also extracted at the same time as the radiosonde observation. Our WRF simulation results have eight "observations" per day.

49. L482-488: This part of the discussion should be extended. For example, for wind energy, is it really LLJs up to 3 km that are of interest?

Yes, thanks for your interest in wind energy again here, as you suggested before, I should add some explanations about how LLJs affect wind power production in Section 1. So now in this revised version, the introduction part will offer a description of LLJs and wind turbine working. Therefore, when readers get here, they can recall the significance of LLJ for the field they are interested in.

**Reviewer #2:**

Minor Comments:
50. Line 49, density in time only or density in time and space?

According to the previous description before this sentence, this should specifically refer to time density. Therefore, references to the shortcomings of spatial density have been mentioned in later words. To avoid being

misunderstood, I have changed the original text here to "time density". Thanks for the correction.

51. Lines 58-59, "perform more extensive measurements" to "provide more comprehensive variables through assimilation."

    Thank you! I have corrected this part in the manuscript as you suggested.

52. Lines 82-83, the authors should emphasize the theory was proposed to explain the diurnal cycle of the Great Plains LLJ by put the phrase in the beginning of the sentence.

    Yes, I have emphasized this theory explains the diurnal cycle at the beginning of the sentence. So now the text must be like "In explaining the diurnal cycle feature of the Great Plains LLJ, the inertial oscillation theory proposed by Blackadar (1957) and Stensrud (1996) suggests that the LLJ is related to the friction change in the boundary layer." You can refer to this in the revised manuscript.

53. Lines 97-100 I suggest the authors to write more concisely and directly by changing it to "In this study, we utilize the 4-km convection-permitting WRF simulation (Liu et al., 2017) to compile a comprehensive LLJ climatology across North America and investigate the features of major LLJ systems in the region with improved spatial and temporal resolutions."

    Sure, we have revised the original texts to enhance clarity and conciseness as suggested. The sentence now reads: "In this study, we utilize the 4-km convection-permitting WRF simulation (Liu et al., 2017) to compile a comprehensive LLJ climatology across North America and investigate the features of major LLJ systems in the region with improved spatial and temporal resolutions" We believe this change improves the manuscript and appreciate your guidance on this matter.

54. Lines 106-114, the length of the simulation should be discussed here and why the forcing reanalysis is chosen as ERA-Interim.

    Sure, I just simply mentioned the reason for choosing this length of simulation in the sentence: "... Considering the computational cost for high-resolution modeling, this simulation period spans from 1st October 2000 to 30th September 2013..."

    As to the input reanalysis dataset choice, this WRF simulation was conducted in 2017 when ERA-Interim was still popular in many studies. Since there is so much experience in ERA-Interim application, this convection-permitting

simulation should also be accepted to analyze the LLJs. But the ERA5 dataset is under our consideration in the future as well.

55. Lines 118-120, may consider change "but" to an alternative more suitable for the two parts.

    Thanks for your reminder, I have changed the whole sentence to:
    "In this study, the planetary boundary layer scheme is retained. Nonetheless, it should be noted that this would introduce uncertainties to the simulation in the vertical direction, especially in regions with complex topography."

56. Lines 126-135, here please justify the categorization of LLJs into northerly and southerly instead of in the direction of west-east. Is it because the geographic contrast (coast, mountains) in the mid-latitude North America mostly north-south oriented? Do LLJs induced by cyclones also have preferred orientation in the N-S instead of W-E?

    Well, in this part, the LLJ classification is based on the study of Walter et al. (2008) and Doubler et al. (2015), and it considers the meridional LLJ for heat and water vapor transport for other research. So, this N-S classification is common, and results would be more comparable if using the same standard in this study. The geographic contrast you mentioned can be one factor of LLJ formations, but it is not the actual reason we chose this wind direction in the paper. Thanks

57. Line 450, change "Convection-permitting simulations can also capture the LLJs that were barely detected previously" to "The convection-permitting simulation can also capture LLJs that were poorly detected previously using coarser resolution modeling and observational datasets."

    Sure, I have revised it as you suggested.

58. Lines 489-496, Figure 15-16, though related to LLJs, seem to delve to the effects of LLJs on precipitation instead of characteristics of LLJs like in the remainder of the article.

    Thank you for your inquiry regarding the discussion of the impact of low-level jets (LLJ) on precipitation in the manuscript. This content was added to address a previous reviewer's comments in the submission to another journal. The previous reviewer suggested that the role of LLJs near Quebec seems to be less influential and less studied, so the whole manuscript looks less significant. We believed that this addition provided valuable context and clarification that could be beneficial for readers in understanding that

Quebec N-LLJs are also important to understand the local winter precipitation and climate, offering the reference to meteorologists in this region. You mentioned this part is more suitable in a companion paper for an in-depth investigation, I think this is a good suggestion and would consider it in future work. Thank you!
* * *
I would be happy to make any further changes that may be required.

Thank you for your consideration and suggestions.

Sincerely,

Yanping Li

---

## Author Response (AR2)

Dear editor,

Thank you for your detailed feedback on our manuscript revisions. We greatly appreciate the constructive suggestions and clarifications provided by you and the reviewers.

So, in this round of revision, we have carefully considered and responded to each comment from the reviewers. Here is the response below:

**Reviewer #1:**

Major Comments:
1. Title and throughout the manuscript: Since only 13 years of data are used for the analysis, I think the authors should be very careful calling the work a climatology (which you also mention, line 509-510). The standard normal is to use 30 years for a climatology. All places in the text where climate/climatology/climatological is mentioned should be carefully checked and changed appropriately.

   Thanks for your suggestions, firstly I should explain the purpose of this setup: The 13-year simulation may not fully suffice for comprehensive climatological analysis, it represents a pragmatic approach given the substantial computational demands of convection-permitting simulations across a broad study area. The challenge of executing high-resolution simulations over several decades renders 13 years a balanced compromise. Liu et al. (2017, **Continental-scale convection-permitting modeling of the current and future climate of North America**, DOI: 10.1007/s00382-016-3327-9) have applied the 13-year simulation to investigate the current and future climate in North America and also offered convincing results. We also explained this in the Methodology part, which you can refer to lines 120-122.

   But we acknowledge that "13 years" is commonly believed not enough to analyze the "climatology", so we accept your advice here and change the title to "Investigation of the characteristics of low-level jets over North America in a convection-permitting WRF simulation". Meanwhile, we reduced the text where climate/climatology/climatological is mentioned as possible in the manuscript.

Thinking of this limitation in the discussion, we will extend our research to longer-term, large-domain regional simulations as advancements in technology enhance computational feasibility in the future.

2. Why did you use ERA-Interim? ERA5, the successor of ERA-Interim, is state-of-the-art and has been around for many years now. It also has higher spatial and temporal resolution than ERA-Interim, which is important for your study. It's not clear why you don't perform the downscaling based on ERA5. The choice of using ERA-Interim instead of ERA5 needs strong motivation.

Yes, indeed, ERA5 is a popular and powerful input data currently. In this study, we used the 4-km WRF simulation dataset generated by Liu et al. (2017), which used ERA-Interim to drive the model while ERA5 was unavailable then. There are many studies based on it. ERA-Interim has been accumulating a wealth of application cases and experiences. Of course, the simulation based on ERA5 will be the focus of our future work, but here, according to the results part, our ERA-Interim-based work is still robust.

3. The analysis of northerly and southerly LLJs (N-LLJs and S-LLJs) is very interesting and important. However, easterly LLJs (E-LLJ) and westerly LLJs (W-LLJ) are completely neglected. The bins should be +/-45° around the four cardinal wind direction. Including all wind directions in the analysis is needed to strengthen this manuscript and is crucial for publication. Further comments on this topic are given below.

This classification criterion, based on the study of Walter et al. (2008) and Doubler et al. (2015), is commonly used in relevant research , and it considers the meridional LLJ for heat and water vapor transport . In this paper, we compared our results with previous studies to examine if the LLJ capture is robust. Therefore, the results are more comparable when using the same standard in this study. For the details of the standard: southerly LLJs (S-LLJs), the jet-core wind direction is between 113° and 247°; for northerly LLJs (N-LLJs), the jet-core direction is between 293° and 67°. This means most (about three-fourths) of the wind directions have been considered in our study by this criterion.

4. L468-482 and Figures 15 and 16: This comes across to me as a rather different topic from the rest of the work. I strongly suggest to cut it from this study. It is interesting but should be a paper on its own.

Thank you for your feedback on our manuscript. In fact, the purpose of these two figures is to supplement the valuable context and clarification that the Quebec N-LLJs are also important to understand local winter precipitation and climate, offering the reference to meteorologists in this region. But we

agree with your suggestions here, because the description in the Discussion part can make the whole paper lose the main logic line. So we have removed this section. We could plan to explore these aspects in future work.

Minor Comments:

1. Title: I think a title that better describes the uniqueness of the work is preferable. A suggestion is Investigation of the occurrence of low-level jets in different wind directions over North America

   Based on the discussion about "13 years" in the response to Major comment#1, we have revised the title into "Investigation of the characteristics of low-level jets over North America in a convection-permitting WRF simulation".

2. L26-27: It sounds like an LLJ is always present in the lower atmosphere, which is maybe not always the case? Please rephrase if LLJs only appear sometimes or even most of the time.

   Thanks for your correction. This sentence has been revised to "A low-level jet (LLJ) is described as the fast-moving air ribbon located in the lower atmosphere most of the time (Bonner, 1968; Rife et al., 2010)."

3. L29: A lot of research has also been performed to study the offshore LLJ. This should also be mentioned, with appropriate references.

   References added: "... Besides, not only these in-land LLJs are investigated, but also offshore coastal LLJs such as the California LLJs (Parish, 2000) and North African Coastal LLJ (Soares et al., 2018) ..." Thanks! (Lines 29-30)

4. L31: Reference needed for "but its width can reach several hundred kilometers"

   Sorry, it was a typo. The description should be "but its length can reach several hundred kilometers", rather than "width". (lines 31-32)

5. L33-35: Please extend and briefly explain how LLJs affect wind power, air pollution, and urban heat islands.

   More detail added:
   "Meanwhile, researchers have long been interested in investigating their features, because LLJs also affect various processes such as wind power development, air pollution transportation, and urban heat islands: the wind turbines would be influenced by positive wind shear and downward entrainment from the LLJs above them, assisting in extracting energy from

the strong wind belt inside LLJs (Gadde and Stevens 2021; Ma et al., 2022). LLJ-related horizontal transportation is beneficial to pollutant removal (Sullivan et al. 2017). The LLJs can enhance the turbulent mixing in the boundary layer thereby decreasing the atmospheric stability, helping pollution diffusion, and weakening urban heat island intensity (Hu et al., 2013)."
(lines 34-40)

6. L35: A great part of the LLJ research has focused on only the lowest 300, 500, or 1,000 m of the atmosphere. It should be mentioned already here in the Introduction that wind profiles all the way up to 3,000 m are assessed for LLJs in this study.

   In fact, according to what you mentioned, the research on LLJs is concentrated below 1 km because the core of most LLJs is at this location. But the standard definition of 3 km here means that I need to find the two minimums of the wind profile within a height of 3km to identify a "nose". In other words, the minimum wind speed above the jet core needs to be found within a height of 3km. Actually, the core heights of the low-altitude jets found in this study are all below 1km, which is consistent with other studies.

7. L40-46: This part feels a bit out of place to me, and should be removed as you are not using any observational data in your study.

   Thanks. In my opinion, the application of observation data described here mainly emphasizes the advantages of model data in space and time, so I think mentioning these in the introduction can make a good comparison before and after, and at the same time form a progressive relationship with the following text.

8. L55: Also lidar measurements, which can be used to study LLJ in the lowest approximately 300 m of the atmosphere, should be mentioned here.

   Yeah, I have mentioned lidar as you suggested:
   "...Although observation platforms such as radar, PECAN, or lidar which investigate the atmosphere as low as 300 m, can compensate to some extent for this lack of observational data. as well as lidar that investigates the atmosphere as low as 300 m, these approaches are still limited by the spatial coverage of their measurement platforms (Smith et al., 2019)."
   (lines 58-61)

9. L57-59: "Reanalysis data [...] perform more extensive measurements" Reanalyses don't perform measurements, so I'm not quite sure what you mean here. Please reconsider the formulation of this sentence.

Thanks for pointing this out. I have revised the text to:
"Reanalysis data have relatively better spatial and temporal coverage than rawinsonde measurements, incorporate observations into the preliminary model simulations, provide more comprehensive variables through assimilation, and contain broader domains." (lines 63-65)

10. L61: "previously unknown jets" Please add information about which these unknown jets are.

Sure, I have listed the unknown LLJs in the sentence, they are "Tarim nocturnal LLJ in northwest China, Ethiopia nocturnal LLJ, and Namibia–Angola nocturnal LLJ." (lines 67-68)

11. L77: Add a reference for the "shows promise" statement

Actually, the following reference about Du and Chen (2019) is an example of using the convection-permitting model, they highlighted the coastal terrain. Maybe "shows promise" is not an accurate description here, so I replaced "promise" with "ability". (line 84)

12. L80-81: References to studies showing that models with higher resolution show more precise results regarding LLJs should be added here.

This sentence is actually the extension of the previous reference about Du and Chen (2019), so I did not add the literature here.

13. L82-96: This description of the formation mechanisms should be presented earlier in the Introduction (perhaps after L35?). Further, coastal LLJs, cold front LLJs, and typical formation processes of offshore LLJs should be explained here.

Because the literature review on the LLJ mechanism is very long in the introduction, I think the entire article will look a bit confusing if it is placed earlier. My logic is still to first describe the observations and simulations of LLJ, and then describe the mechanism. This order will also be consistent with the order in the results section below. Meanwhile, the purpose of this paper is not a pure literature review of LLJ classification and formation, so I do not tend to explain every aspect in the introduction part. Thank you for your advice.

14. L107: Did you use any nesting in your downscaling? It should be stated here as well.

Sure, the simulation did not use nesting, I have revised the sentence in section 2 (lines 111-113). Thanks!

15. L109: Please change throughout the entire manuscript and refer to your work as "downscaling" of ERA-Interim rather than a "simulation".

Thanks, I think the dynamic downscaling of ERA-Interim data can be described as "simulation", because the whole procedure was just inputting coarse large-scale ERA-Interim and conducting the simulation on the smaller grid space. It utilized the physical processes of numerical weather prediction. So "simulation" word in this paper is accurate.

16. L110-112: "five layers under 500-m height and nine layers under 1 km are outputted above ground level, which means the WRF has the good ability to capture the LLJs occurring in the boundary layer". No! This statement is not correct, because a minimum of three levels are required to identify an LLJ and the height of the boundary layer can in some circumstances be very low (even below 100 m). On top of that LLJs can also be very narrow in height, in some cases the core is only 50-100 m in its vertical extent.

Thanks, here is just for a comparison, because compared with other modeling data, this density of vertical layer can really capture more LLJs in the lower atmosphere as you said. But just relatively more, so maybe in the absolute standard, 4km WRF is not good, but maybe it is "better ability than GCMs/RCMs". So I revised the description here. (lines 115-117)

17. L113: What was the vertical resolution of ERA-Interim used as input data for the WRF downscaling? Please add this information.

Sure, in this paragraph I have added "…, the vertical layer depth of inputted ERA-Interim data under 5 km is about 0.3-1.4 km (Hoffmann & Spang, 2022)."
(lines 120-121)

18. L134: At which height level is the wind direction used for classifying the LLJ as S-LLJ or N-LLJ? Or is it the wind direction at the core of the LLJ? Please clarify.

Yes, the LLJ is classified by the jet core wind direction, I have clarified this in the manuscript now (line 144), thank you!

19. L142-157: This analysis and Fig. 2 is not relevant and the connection to LLJs has not been motivated. Following the major comment above that the study should focus on LLJs from the four wind directions (N, E, S, W), maps

showing the relative occurrence of those wind directions for the different seasons should be included instead.

Even though Fig. 2 is not really relevant to the results of LLJs, it is able to be shown as proof of the ability of WRF simulation, by illustrating the large-scale circulation over North America. On the other hand, based on the study of Walter et al. (2008) and Doubler et al. (2015), and it considers the meridional LLJ for heat and water vapor transport for other research, this Northerly and Southerly classification is common. Under the same standard, it is easier to have the comparison of models' performance. But I am grateful for your suggestion!

20. Fig 3.: I suggest a 4x4 panels figure showing the occurrence of LLJs in different wind directions as the rows (N, E, S, W) and different seasons as the columns (DJF, MAM, JJA, SON). This would simplify comparisons between the occurrence in different wind directions and in different seasons. For Fig.3, white color could be used for the lowest percentages (0-1%) and a discretized colorbar (as in Fig. A1 in the Appendix) would increase the readability of the plots. Further, a 4x4 panel figure showing the average LLJ core height (fore sites with at least 5 or 10% relative occurrence) in different seasons and in different wind directions would be valuable and could strengthen the discussion.

Similarly, I have already clarified the reason why I chose only two directions of LLJ-core wind in response to the last comment. Due to the page size scale limitation, I hope to show clearer details when I describe the S-LLJ and N-LLJ, respectively. So I divided this part into sections 3.2.1 and 3.2.2, and made Figures 3 and 4 reach the max width.

As to using the discretized color bar, I have regenerated the discretized color bars in Figure 3-8 (in section 3) as you suggested and applied the same colors in Figure A1 (Appendix). Meanwhile, the white color has been used to represent the level of "hardly occur". I believe the figures are more readable now. Thanks!

And I agree that the jet-core height can be a valuable topic for LLJs, but this paper was planned to focus on the occurrence distribution and physical mechanisms, we think the core height could be a promising direction for future work.

21. L210: "British vf Canada", what is "vf"?

Sorry, it is a typo, the original word should be "British Columbia, Canada". Thanks for correction. (line 220)

22. L215-219: This validation doesn't hold. If ERA5 is trusted as the best description of LLJs, why don't you use ERA5 in the first place? Validation of the WRF downscaling has to be against observations (e.g., radio soundings), potentially showing superior performance of the WRF downscaling as compared to ERA5 or ERA-Interim. Following this remark, the Appendix should be removed.

    Thanks, the purpose of the appendix is to prove the performance of WRF simulation in capturing basic LLJs, rather than simply comparing the advantage of the convection-permitting model. Indeed, ERA5 is a much better dataset than others, the reason why we don't use ERA5 in the first place is actually the climate simulation can have future results, and the future change in LLJs is also under consideration in future plans. So we applied the 4km WRF model, and examined if WRF can capture basic LLJs in the appendix, and then in the future work we will discuss the impact of climate change. This is the logical flow of our project.

23. L222: In summer, yes, but it is not true that S-LLJs appear most frequently in winter, so please rephrase this sentence.

    Sure, I have rephrased the sentence as:
    "To show the diurnal features of the LLJs, we selected summer and winter as the representative seasons because S-LLJs and N-LLJs occur most frequently in these seasons, respectively. Below, the descriptions are divided into N-LLJs and S-LLJs. " (lines 231-233)

24. L226: Please also state in the text what 21 UTC is in the local time zone for California.

    Thanks for the reminder, 21 UTC is 1 pm in the local time zone for California, I have added the statement in the manuscript as:
    "... from 21 UTC (1 pm LST in California), the low-level jet begins to develop, with a N-LLJ frequency of >30%..." (line 237)

25. L229-230: It is not possible for me to spot the Hudson Bay N-LLJ. Remove this sentence as you are not describing other LLJs that appear only approximately 5% of the time.

    Yes, I removed this description about an insignificant LLJ in the revised paper, thanks.

26. L233: I think Hudson Bay Lowlands is a more common name than Hudson Bay Plain, so please change.

Sure, the "hudson bay plain" has been replaced with "hudson bay lowlands".

27. L235: Again, please state what 18 UTC is in local time.

Yes, because Hudson Bay Lowlands, Quebec Labrador Plateau, and the Appalachians are all in the Eastern Time Zone, so the local time is 5 hours behind UTC. I have stated the correct local time in the paragraph. (line 245)

28. L258: Maybe change "significant" to "pronounced" instead? Unless you performed some significance testing?

Sure, the word "significant" has been replaced as advice, thanks. (line 268)

29. L266-293: This part should be moved to form a new Theory section, placed between Sects. 1 and 2.

Thank you very much for this valuable suggestion. After consideration, we have decided to maintain the current structure. I think the related mechanism theory in its current location can give a more intuitive explanation for the subsequent results part. If I put the theory part after Section 1, it may be hard for readers to refer to it when they scan to Section 4. Furthermore, I am also concerned that reorganizing the sections might disrupt this balance among sections, resulting in previous sections being lengthy or information-dense, potentially affecting readability. But I am very grateful for your concern here!

30. L297: Please refer to the figures where this can be seen.

Sure, Figure 7 should be referred to here, I have added relevant words, thanks.

31. Figures 9, 11, and 13, panels a and b: Plot anomalies in panels a and b instead of the average fields, to show how much the pressure or geopotential height differs compared to normal in cases when an LLJ is present.

The anomaly field will indeed provide more significant patterns than the climate mean field, but here I still hope to retain the climate mean field, because then I can refer to more large-scale climate systems for helping to analyze the results. In addition to discussing the relationship between LLJs and large-scale circulation, we can continue to use these figures as performance proof of WRF simulation. Thanks for your suggestion here.

32. Figures 9, 11, and 13, panel c: Please also show the location of the selected site along the x-axis.

    I have used one vertical line in subplot c to represent the zonal location of the selected jet core.

33. Figures 9, 11, and 13, panel d: Add confidence interval to show if the diurnal cycle really is significant.

    I have used the gray shading to show the confidence interval of the diurnal cycle plot.

34. Figures 9, 11, and 13: In addition to panel d, which is very interesting, also include plots showing the diurnal cycle of core speed and core height. Also, in panel d, let the line extend over the left and right edges to show the changes there (i.e., in Fig 9d, 23 LST and 2 LST should be "connected" with a line).

    Yes, now in subplot d of Figures 9, 11, and 13, the lines are connected by extension.

35. Fig 9: Please add in the caption that this is for the JJA season.

    Thanks, I have added the season information in the caption.

36. L318-319: Why not use the actual core height at every given time step? It might vary drastically from the most frequent core height.

    Thank you for your question. In the manuscript, the position of the "jet-core" I explained refers to the position in the horizontal direction where LLJ occurs most frequently, which is the point in Figures 9a and 9b. And the jet-core height is indeed determined at each time step. So it is calculated as you said in the question. But obviously, something is confusing in my manuscript, so I've added "horizontal" to this sentence to make it clear. (line 329)

37. L326: Please clarify by adding "where i is the index of the grid point at point-a".

    Sure, thanks for your clarification, I have finished for this! (lines 337-338)

38. Figures 10, 12, and 14, panels abc: Please scale so that 5 m/s on the x-axis has the same size as 5 m/s on the y-axis.

I appreciate your suggestion. So I have rescaled the x-axis tick size of Figures 10 and 12, to make them look the same as the y-axis. You can find this modification in sections 4.1 and 4.2. However, it is significant that the meridional components of the California coastal N-LLJ outpace their zonal counterparts much (Section 4.3, Figure 14). Matching the y-axis ticks to the x-axis would elongate the geostrophic wind vectors in these scenarios, aligning them almost parallel to the y-axis. At the selected local time, the ageostrophic wind vectors would appear negligible next to the geostrophic ones, which is obscuring the intended results. Thus, I only maintain the current y-axis tick scaling in Figure 14. Thank you for your understanding!

39. Figures 10, 12 and 14: Please adjust the colours to account for people with color vision deficiency, red and green is unfortunately not the best combination.

   Sure, I have used red and royal blue to make the contrast of different winds in this kind of figures. Thanks!

40. L353 (and elsewhere): Please note the N-LLJ is often used as an abbreviation for nocturnal LLJ. I suggest switching to write LLJ(N) (and similar for the other wind directions) which would simplify for the reader.

   Thanks for the reminder. In some papers, the writers used N-LLJ to represent the nocturnal LLJ, but it can also be the abbreviation of Northerly LLJ, for example, in the paper named "A Long-Term Climatology of Southerly and Northerly Low-Level Jets for the Central United States" (Walters et al., 2008). I think if I explain clearly at the beginning, it would be okay.

41. L359: Add "relatively" before "warm sea".

   Sure, thanks! I have finished for this! (line 371)

42. General for Sect. 4.1-4.3: Please also comment on the stability of the atmosphere when LLJs are occurring (panels c in Figures 9, 11, and 13)

   Sure, I have added the description in sections 4.2 (lines 374-379) and 4.3 (lines 424-433) to discuss the atmospheric stability by isentropic lines. And try to explain the different wind speeds of three cases by this stability theory.

43. Figures 11-14: Please add full captions for all these figures (even if it is repetitive it is easier for the reader than having to scroll back and forth).

   I have finished revisions you suggested here, thanks.

44. L392: How is it similar? It's not clear to me.

    Well, here the word "similar" means California coastal N-LLJ is also located upon the contrast between sea and land, even though it is offshore but Quebec N-LLJ is onshore mostly. But they may have a "similar" mechanism of formation, this is what I want to state in this sentence.

45. L403: "Compared with the Quebec LLJ, California's maximum central wind speed exceeds 20 m s-1". This sentence should be rephrased as it is not a comparison.

    I have rephrased the sentence into "The maximum central wind speed of California coastal LLJ exceeds 20 m s-1, whereas Quebec N-LLJ's max core wind is only about 14 m s-1." (lines 426-427)

46. L408: The fact that California's LLJ occurs frequently cannot be the reason for that the diurnal signal is weak.

    Here I just wanted to present that the California N-LLJ occurs much frequently at each time step, and then the diurnal cycle did not look obvious in Figure 13d. There is no causation between these two descriptions. But this sentence is weird and confusing, I have modified it to "California's LLJ occurs frequently at each time step, its diurnal signal is weaker compared, for example, to the signal in the Great Plain S-LLJ". Thanks for correction. (lines 433-434)

47. L433: The model level output from ERA5 actually has a better vertical resolution than the WRF output you are using, so this sentence should be expressed in a more careful way.

    I understand, so I removed the phrase "...especially in the vertical direction, ..." to make the whole sentence more accurate. Thanks! (lines 465)

48. L435-436: It is not clear how using a convection-permitting model actually gives better resolved LLJs. Since LLJs tend to appear in stable conditions (where the convection-permitting features shouldn't be important), this statement has to be stronger motivated.

    Thanks for your suggestion, so "more accurate" in the original manuscript may not be that motivated, I used "relatively more comprehensive" to replace this phrase. At least considering the spatial resolution and temporal density, the convection-permitting model can really give "more comprehensive" results. (line 468)

49. L447-449: The numbers presented here are interesting, but since you did not study the exact same time period it is difficult to draw any conclusions based on this. A comment about this should be added here.

Thank you for your comments. Firstly, during the comparison, I have tried to keep the observation time consistent, because the radiosonde data only has two observations a day, and the occurrence frequency I mentioned here is also extracted at the same time point as the radiosonde observation. Our WRF simulation results have eight "observations" per day.

And then in the whole period, I admit that the data presented in this study are not from the same period. However, for climatological research, even data over different periods can still provide valuable insights and references. This is because climatology focuses on long-term trends and patterns that often transcend specific time frames. Therefore, despite the different periods, a comparison of these data still reveals some key features. But in the revised draft, I have emphasized that the periods are not the same to offer more accurate reference. (lines 483-484)

50. L482-488: This part of the discussion should be extended. For example, for wind energy, is it really LLJs up to 3 km that are of interest?

Yes, thanks for your interest in wind energy again here, as you suggested before, I should add some explanations about how LLJs affect wind power production in Section 1. So now in this revised version, the introduction part will offer a description of LLJs and wind turbine working. Therefore, when readers get here, they can recall the significance of LLJ for the field they are interested in. (lines 36-38)

**Reviewer #2:**

Major Comments:
1. The paper explores the climatology of low-level jets (LLJs) over North America, utilizing a high-resolution (4 km) convection-permitting Weather Research Forecasting (WRF) simulation. Investigating LLJ characteristics and their impact on extreme precipitation events is a valuable contribution to climate research but seems more suitable to be in a companion paper for a in-depth investigation.

Thank you for your constructive feedback on our manuscript. We initially included the section on low-level jets (LLJs) and their impact on local winter

precipitation and climate to provide readers with valuable context and clarification on the significance of Quebec N-LLJs. However, we agree with your suggestion, and recognizing that this might have diverted focus from the core objectives of the paper, we have **removed** this section. We could plan to explore these aspects in future work and dedicate the necessary space to detail their implications for meteorology in the region comprehensively. This revision will help maintain the clarity and focus of the current work. So you may refer to the revision in the Discussion part of the draft.

2. The use of a 13-year simulation at 4km on a continental scale period provides substantial data for analysis, allowing for the examination of LLJ occurrences on a seasonal, diurnal, and regional scale over North America, however it is still shorter than the usual definition of climatology (30 years). I guess this is due to the cost of computation and should be included in the data/methodology section to justify the use of a shorter period for climatology.

   Please allow me to explain first: Actually, because conducting high-resolution simulations over several decades is much more time-consuming, and given the current constraints of computational resources, a 13-year period is a pragmatic compromise. We admit that it is shorter than the normal and comprehensive climatology study, but it is also very practical considering the computing over the large areas with fine grids. We have added the related description in the Methodology part as you suggested, so you can refer to this in lines 121-123. Thank you!

   Moreover, we admit that "13 years" is commonly believed not enough to analyze the "climatology", so after consideration, we have changed the title to "Investigation of the characteristics of low-level jets over North America in a convection-permitting WRF simulation". And reduced text where climate/climatology/climatological is mentioned as possible in the manuscript. Now we think the draft is more accurate in this point.

   We recognize this limitation in our study, and plan to conduct longer and larger-scale regional modeling as future technological advances make such efforts more computationally manageable. We have added some revisions in lines 120-122, thanks for your feedback here!

3. The identification of well-known large-scale LLJs, such as the southerly Great Plains LLJ and the summer northerly California coastal LLJ, confirms the credibility of the simulation results. Additionally, the discovery of the Quebec northerly LLJ, which has received less attention previously, adds novelty to the study.

We are grateful to the reviewers for their recognition of our work on the identification and analysis of large-scale low-level jets (LLJs). Your positive comments are of great significance for us to continue our in-depth research. We will continue our efforts to ensure this novel discovery is fully researched and reported. Thank you again for your valuable comments and suggestions.

4. The analysis of weaker and smaller-scale LLJs in complex terrain regions, like the foothills of the Rocky and Appalachian Mountains, is interesting. These findings shed light on LLJ characteristics in regions with distinct topography, which can be crucial for localized weather events. It also shows the advantage of using a finer resolution in regional climate models.

Thank you for your positive feedback on our study's section regarding the low-level jets (LLJs) in complex mountain terrains. We are pleased to find out that the use of high-resolution models is crucial for detailed analysis, and we agree with you that this result can enhance the understanding of terrain influences on local weather. This can be further extended in our research, to elaborate on these small-scale systems and related weather system modeling. We appreciate your insights here and will continue to refine our approach and contribute to the field!

Minor Comments:
1. While the abstract provides an overview of the study, it would be helpful to include specific findings or results to give readers a better understanding of the research's significance. Right now the abstract includes the identification of several LLJs without much details.

Thanks, we have revised the abstract and included the information on LLJ's seasonal/diurnal variations, different formation mechanisms with various LLJs. Now we believe the abstract can show more helpful instructions to readers. You can refer to the change in line 10-23 of the new draft!

2. It would be beneficial to discuss the limitations of the WRF simulation and the potential impact on the study results, considering factors such as model biases and uncertainties.

Sure, I have discussed the relevant limitations of this simulation in the last paragraph of Section 5 (lines 519-522)

3. Line 49, density in time only or density in time and space?

According to the previous description before this sentence, this should specifically refer to time density. Therefore, references to the shortcomings of spatial density have been mentioned in later words. To avoid being misunderstood, I have changed the original text here to "time density". Thanks for the correction. (line 54)

4. Lines 58-59, "perform more extensive measurements" to "provide more comprehensive variables through assimilation."

   Thank you! I have corrected this part in the manuscript as you suggested. (lines 64-65)

5. Lines 82-83, the authors should emphasize the theory was proposed to explain the diurnal cycle of the Great Plains LLJ by put the phrase in the beginning of the sentence.

   Yes, I have emphasized this theory explains the diurnal cycle at the beginning of the sentence. So now the text must be like "In explaining the diurnal cycle feature of the Great Plains LLJ, the inertial oscillation theory proposed by Blackadar (1957) and Stensrud (1996) suggests that the LLJ is related to the friction change in the boundary layer." You can refer to this in the revised manuscript. (lines 89-91)

6. Lines 97-100 I suggest the authors to write more concisely and directly by changing it to "In this study, we utilize the 4-km convection-permitting WRF simulation (Liu et al., 2017) to compile a comprehensive LLJ climatology across North America and investigate the features of major LLJ systems in the region with improved spatial and temporal resolutions."

   Sure, we have revised the original texts to enhance clarity and conciseness as suggested. The sentence now reads: "In this study, we utilize the 4-km convection-permitting WRF simulation (Liu et al., 2017) to analyze the features of low-level jet systems across North America, improving the spatial and temporal resolutions" We believe this change improves the manuscript and appreciate your guidance on this matter. (lines 104-105)

7. Lines 106-114, the length of the simulation should be discussed here and why the forcing reanalysis is chosen as ERA-Interim.

   Sure, I just simply mentioned the reason for choosing this length of simulation in the sentence: "… Considering the computational cost for high-resolution modeling, this simulation period spans from 1st October 2000 to 30th September 2013…" (lines 117-118)

As to the input reanalysis dataset choice, this WRF simulation was conducted in 2017 when ERA-Interim was still popular in many studies. Since there is so much experience in ERA-Interim application, this convection-permitting simulation should also be accepted to analyze the LLJs. But the ERA5 dataset is under our consideration in the future as well.

8.  Lines 118-120, may consider change "but" to an alternative more suitable for the two parts.

    Thanks for your reminder, I have changed the whole sentence to:
    "In this study, the planetary boundary layer scheme is retained. Nonetheless, it should be noted that this would introduce uncertainties to the simulation in the vertical direction, especially in regions with complex topography." (lines 128-130)

9.  Lines 126-135, here please justify the categorization of LLJs into northerly and southerly instead of in the direction of west-east. Is it because the geographic contrast (coast, mountains) in the mid-latitude North America mostly north-south oriented? Do LLJs induced by cyclones also have preferred orientation in the N-S instead of W-E?

    Well, in this part, the LLJ classification is based on the study of Walter et al. (2008) and Doubler et al. (2015), and it considers the meridional LLJ for heat and water vapor transport for other research. So, this N-S classification is common, and results would be more comparable if using the same standard in this study. The geographic contrast you mentioned can be one factor of LLJ formations, but it is not the actual reason we chose this wind direction in the paper. Thanks

10. Line 450, change "Convection-permitting simulations can also capture the LLJs that were barely detected previously" to "The convection-permitting simulation can also capture LLJs that were poorly detected previously using coarser resolution modeling and observational datasets." (lines 485-486)

    Sure, I have revised it as you suggested.

11. Lines 489-496, Figure 15-16, though related to LLJs, seem to delve to the effects of LLJs on precipitation instead of characteristics of LLJs like in the remainder of the article.

    Thank you for your feedback, as I mentioned in the response to your first main comments, the description of precipitation here can really make the topic lose focus. So we accept your suggestion, remove all the associated texts in this paper, and may discuss them again in the future job.
* * *
I would be happy to make any further changes that may be required.

Thank you for your consideration and suggestions.

Sincerely,

Yanping Li

---

## Author Response (AR3)

Dear editor,

Thanks for your suggestions to my second round of revision, I am very glad to see your approval of our modifications. Regarding the two questions you raised, here are our responses:

1.  Modification related to Reviewer #1-main comments #2.

    Thanks, I have modified the draft, you can refer to the changes in lines 121-122 and lines 527-530.

2.  Questioning about wind directions to define LLJs.

    Regarding the wind direction chosen when defining LLJ. I think I need to clarify in the manuscript that the research on LLJs in North America is based on the two directions I mentioned, which are related to the shape of the main terrain of North America, the direction of the coastline, and the large-scale circulation system. So, in the following Section 4, I also conducted a relevant mechanism analysis. Meanwhile, the study of Walter et al. (2008) and Doubler et al. (2015), did not mean to suggest considering LLJs in the N-S direction only. For the North American continent, the Great Plains S-LLJ and California N-LLJ have attracted much attention, then if the later research could focus on the same kind of LLJs, it would be more beneficial to comparison. Besides this, the meridional water vapor, heat, and momentum transport are relatively important to the North American climate system. Therefore, focusing on the N-S direction is more appropriate for research in North America.

    LLJs in the W-E direction also exist in other areas, for example, the West African Westerly Jet (Bing Pu and Kerry H. Cook, 2010, *Dynamics of the West African Westerly Jet*, https://doi.org/10.1175/2010JCLI3648.1). The mechanism for its formation is related to the westward extension of the continental thermal low pressure over Africa in summer, which has a very certain local characteristic. And this zonal LLJ is helpful to the water vapor transport in West Africa, so it has attracted the interest of local researchers. Thus, the choice of investigation should normally match the local feature. I adopted part of this answer for the draft revision, you can refer to them in lines 143-148.
* * *
Thank you for your consideration and suggestions. I am looking forward to your further responses.

Sincerely,

Yanping Li